# REASONAUDIO: SEMANTIC REASONING AND TEMPORAL SYNCHRONY IN VIDEO–TEXT-TO-AUDIO GENERATION

## ABSTRACT

The rapid advancement of video-text-to-audio (VT2A) diffusion models has enabled unprecedented audio generation conditioned on video and text, yet two major challenges remain: following complex semantic descriptions and achieving robust audio–visual synchronization. In this work, we propose ReasonAudio, an MLLM-empowered flow-matching generative model with stronger semantic and robust temporal alignment. To enhance semantic understanding, we 1) address the scarcity of semantically rich tri-modal (video–text–audio) annotations by constructing VGGSound-Think, a dataset enriched with acoustic hints and audio–visual relation descriptions, and 2) leverage MLLMs to understand multimodal conditions (video and text) by introducing learnable queries that bridge understanding and generation components. To tackle temporal alignment, we employ preference optimization (Flow-DPO, Flow-RWR) with synchronization feedback, aligning generative models with visual synchrony preferences. Extensive experiments demonstrate that ReasonAudio achieves state-of-the-art performance in VT2A generation, with substantial improvements in both semantic alignment and temporal synchronization. Moreover, evaluations on VGGSound-Think show that our model excels at reasoning over acoustic hints and following descriptions of audio–visual relations (e.g., object interactions and on-/off-screen attribution). The demo page is available at https://ReasonAudio.github.io.

## 1 INTRODUCTION

Deep video–text-to-audio (VT2A) generation aims to synthesize ambient sounds (e.g., rain, river flow) with convincing details conditioned on video and text. Recent advances (Cheng et al., 2025; Liu et al., 2025a) have made progress by adopting multimodal joint training paradigms that condition audio generation on both video and text inputs. Despite these advances, VT2A systems still face two persistent gaps: achieving 1) robust audio–visual temporal alignment; and 2) strong semantic alignment, i.e., reasoning from acoustic hints and following audio-visual relation descriptions.

Effectively encoding conditions (i.e., video and text) for precise semantic alignment remains critical challenges for two reasons: 1) Data scarcity. Text annotations in common audio–visual pairs (e.g., VGGSound (Chen et al., 2020)) are sparse and semantically shallow, limiting the ability to follow descriptions on audio–visual relations—such as object interactions and on-/off-screen attribution. 2) Modeling. Contrastive pretrained encoders (Radford et al., 2021; Elizalde et al., 2023) provide compact, informative features for diffusion models, while their maximum token limit becomes a significant constraint to encode long structured descriptions. Recent approaches (Ge et al., 2024; Sun et al., 2023; Team, 2024) leverage Multimodal Large Language Models (MLLMs) to produce semantically and temporally aligned reasoning instructions for guiding audio diffusion models, while these pipelines introduce substantial complexity due to multi-stage training for the need to bridge LLMs with diffusion backbones.

A second challenge lies in achieving robust audio-visual synchronization (i.e., temporal alignment). Prior works (Wang et al., 2024; Luo et al., 2023) rely on contrastive audio–visual pretraining representations, and MMAudio (Cheng et al., 2025) introduces a conditional synchronization module that leverages high-frame-rate visual features to model temporal relationships. However, automatic

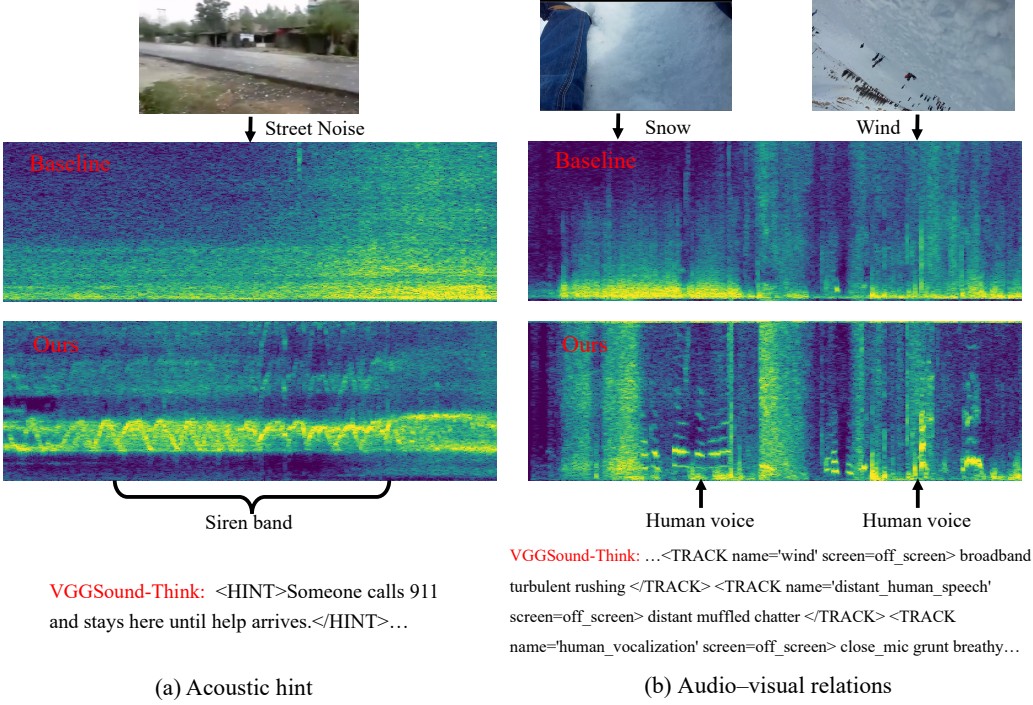

VGGSound-Think: <HINT>Someone calls 911 and stays here until help arrives.</HINT>…

VGGSound-Think: …<TRACK name='wind' screen=off_screen> broadband turbulent rushing </TRACK> <TRACK name='distant_human_speech' screen=off_screen> distant muffled chatter </TRACK> <TRACK name='human_vocalization' screen=off_screen> close_mic grunt breathy…

(a) Acoustic hint

(b) Audio–visual relations

Figure 1: Video-text-to-audio generation by ReasonAudio (bottom) and baseline (top) in VGGSound-Think, showcasing the strong capabilities to reason over acoustic hints (*"someone calls 911 and stays here until help arrives"*) and follow descriptions on audio–visual relations (*off-screen "human speech"*). In contrast, Baseline fails to connect the text to an appropriate acoustic context, and ignores the off-screen sound instructions.

evaluations (Cheng et al., 2025; Liu et al., 2025a) indicate that audio–visual synchrony remains a significant bottleneck: temporal information in learned video representations is often weak, making them difficult to capture robustly without human priors.

In this work, we propose ReasonAudio for video-text-to-audio (V2TA) generation, an MLLM-empowered flow-matching generative model with improved semantic reasoning and robust temporal alignment. To enhance semantic, we 1) construct VGGSound-Think, an audio–visual dataset enriched with acoustic hint (Figure 1(a)) and audio–visual relation descriptions (Figure 1(b)); and 2) leverage the MLLMs to understand multimodal conditions (video and text), where we freeze the MLLMs and utlize the learnable queries to bridge the understanding and generative components. To tackle audio–visual temporal alignment, we leverage preference optimization (Flow-DPO, Flow-RWR) with synchronization feedback, aligning generative models with visual synchrony.

Both subjective and objective evaluations demonstrate that the model achieves state-of-the-art results in VT2A generation with substantial improvements in semantic alignment empowered by MLLMs, and robust temporal synchronization from preference optimization. As shown in Figure 1, our model excells at reasoning over acoustic hint and following descriptions on audio–visual relations (e.g., object and on-/off-screen attribution). Key contributions of the paper include:

• We create VGGSound-Think, which augments VGGSound with semantically rich textual descriptions, including *acoustic hints* and *audio–visual relations*.

• We propose an MLLM-empowered flow-matching generative model to enhance semantic understanding, with learnable queries to bridge the understanding and generative components.

• We employ preference optimization (Flow-DPO, Flow-RWR) using synchronization feedback, aligning generative models with visual-synchrony preferences.

• We achieve state-of-the-art VT2A performance and demonstrate the outperformed reasoning capabilities in qualitative case studies.

## 2 RELATED WORKS

### 2.1 VIDEO-TEXT-TO-AUDIO GENERATION

Video-text-to-audio (VT2A) generation is a multi-modal audio generation task that requires 1) synthesizing realistic high-fidelity audio signals and 2) bridging video/text and audio modalities to ensure cross-modal alignment. Im2Wav (Sheffer & Adi, 2023) explores image-to-audio generation with language models that operate over a hierarchical discrete audio representation obtained from a VQ-VAE-based model. Diff-Foley (Luo et al., 2024) introduces the contrastive audio-video pretraining to align multi-modal features and trains a latent diffusion model for generation. Frieren (Wang et al., 2024) leverages reflow and one-step distillation with guided vector field for audio generation. Recently, MMAudio (Cheng et al., 2025) proposes the joint training paradigm for video-to-audio data scaling and cross-modal understanding, and shows that joint training not only enhances cross-modal performance but also preserves the effectiveness of single-modality generation.

### 2.2 MULTIMODAL LARGE LANGUAGE MODELS

Recently, the community has witnessed efforts to extend the success of multimodal large language model (i.e., MLLM) (Chen et al., 2025b; Shi et al., 2024) to multimodal diffusion generation. Liu et al. (2025a) fine-tunes MLLMs to generate reasoning chains that explicitly capture temporal dependencies and the decomposition of audio editing events. It necessitate tuning LLMs on video understanding and subsequently training the audio generator, naturally posing challenges from LLM overfitting and multi-stage training. MetaQuery (Pan et al., 2025) uses learnable queries to bridge frozen pre-trained MLLMs with pre-trained diffusion models. BLIP-3o (Chen et al., 2025a) leverages the frozen LLM to understand and train the diffusion model to generate semantically rich CLIP image features. In this work, rather than fine-tuning MLLMs in multiple stages, we freeze the LLM and employ learnable queries to bridge the understanding and generative components.

### 2.3 PREFERENCE OPTIMIZATION

There is often a gap between generative models' training objectives and human preference, and thus human feedback has been utilized to align model performance with user intent to improve performance in downstream tasks. DiffusionDPO (Wallace et al., 2024) adapts the Direct Preference Optimization (DPO) (Rafailov et al., 2023) and aligns diffusion models to human preferences by directly optimizing on human comparison data. In the class of flow-matching generative models (Lee et al., 2023; Liu et al., 2022), which predict velocity rather than noise, Liu et al. (2025b) explores direct preference optimization and reward-weighted regression to extend the diffusion-based preference optimization to flow-based generative models. Recently, preference optimization has been applied in audio generation to enhance semantic alignment between the input prompt and output audio. For example, Tango 2 (Majumder et al., 2024) fine-tunes the text-to-audio model using DPO loss on the constructed preference dataset, demonstrating improved audio quality and relevance. In contrast to semantic feedback, we focus on temporal alignment in video-to-audio generation by incorporating synchronization feedback, where the audio-visual synchrony preference has not yet been studied.

## 3 VIDEO-TEXT-TO-AUDIO REASONING DATASET

### 3.1 BACKGROUND

Joint video-text-to-audio (VT2A) generation training paradigm (Liu et al., 2025a; Cheng et al., 2025) emerges and demonstrates the improved audio quality and semantic alignment through cross-modal understanding. However, the text annotations in common audio–visual pairs (e.g., VG-GSound (Chen et al., 2020)) are sparse and semantically shallow, which hinders learning from acoustic hint and following descriptions on audio–visual relation (such as multi-object interactions and on/off-screen sound attribution). Although Liu et al. (2025a) introduce chain-of-thought captions to enhance semantic understanding, their focus is primarily on temporal relations and interactive editing conditions, without considering the acoustic hints or audio-visual relations.

To achieve strong semantic alignment between input conditions and generated audio, it is crucial to provide semantically rich textual descriptions. As illustrated in Figure 2, we introduce VGGSound-

Think, a tri-modal (video–text–audio) dataset that augments VGGSound with 1) acoustic hints and 2) structured audio–visual relation annotations capturing multi-object interactions and on-/off-screen sound attribution.

## 3.2 Foley Reasoning Caption Generation

We generate audio descriptions using GPT-4o (Hurst et al., 2024) which excells in multi-modal understanding and conversations. Each sample is annotated through a structured, step-by-step procedure:

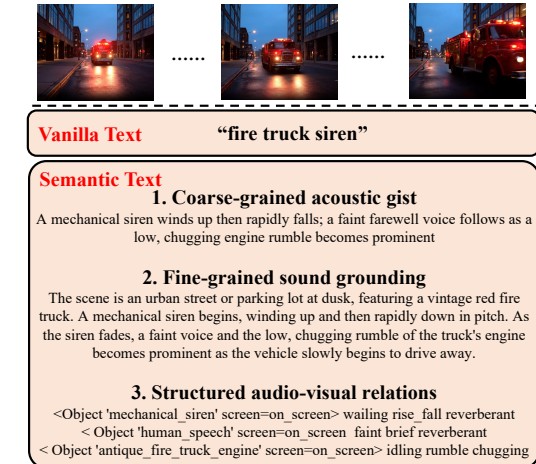

- **Coarse-grained acoustic gist.** For each audio clip, we first provide the high-level hint that summarizes the dominant sound sources suggested by the corresponding video, without explicitly naming the acoustic objects.
- **Fine-grained sound grounding.** We then align the sounds with specific visible objects in the video, refining the gist into a more detailed caption.
- **Structured audio–visual relation annotations.** Finally, for each grounded object we provide descriptive keywords and its audio–visual relations, including on-/off-screen attribution and interactions among objects.

Figure 2: A comparison between vanilla text and semantic text in dataset construction pipeline.

Combined with the associated videos and output audio, these tri-modal (video–text–audio) triplets support semantically rich training and evaluation for VT2A models.

## 3.3 Dataset Validation

After constructing VGGSound-Think, we use metrics (mean pairwise cosine distance, VLM-as-Judge) to evaluate caption diversity and alignment accuracy in Table A.

- **Caption diversity.** We randomly choose 10 video classes (e.g., baby, fireworks), and randomly sample 20 captions and compute the *mean pairwise cosine distance* between their T5 embeddings (higher indicates more diverse phrasing/semantics within the class).
- **Alignment accuracy.** We randomly sample 5% of the full dataset and ask a VLM-as-judge to perform pairwise preference comparisons: given the same video/audio pair and two candidate captions (from VGGSound vs. VGGSound-Think), the judge selects the caption with better audio-visual alignment ("win"). To reduce potential bias from the caption construction process, we use Gemini-3 as an external judge model, rather than the model used for data generation.

More detailed comparisons on MLLM usage and text format have been attached in Appendix D.

| Data | caption diversity (↑) | alignment accuracy (↑) |
|---|---|---|
| VGGSound | 0.51 | 37.4% |
| VGGSound-Think | 0.87 | 62.6% |

Table 1: VGGSound-Think Dataset caption diversity and alignment accuracy validation.

## 4 ReasonAudio

In this section, we overview ReasonAudio and illustrate multimodal large language models (MLLMs) for encoding multimodal conditions (video and text). The MLLM is kept frozen, and

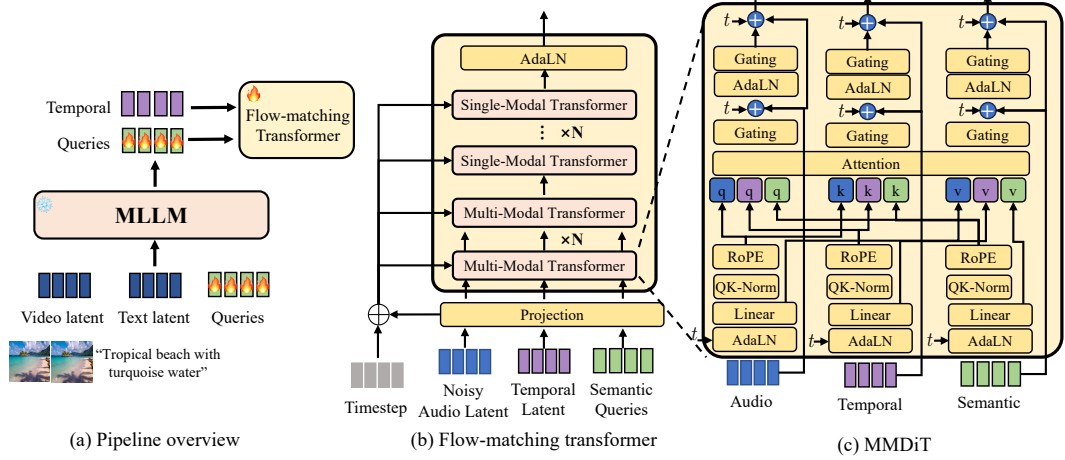

Figure 3: ReasonAudio overview. It adopts a triple-stream MMDiT architecture. The temporal, semantic, and audio representations are respectively provided by a frozen synchformer, learnable queries and a VAE encoder. We use $t$ to denote time embeddings.

learnable queries are introduced to bridge the understanding and generative components. Next, we present the multimodal flow-matching transformer with MM-DiT blocks for effective condition injection. In the following, we employ preference optimization (Flow-DPO, Flow-RWR) using synchronization feedback, aligning generative models with visual-synchrony preferences.

## 4.1 OVERVIEW

As illustrated in Figure 3, ReasonAudio consists of the following main components: 1) MLLMs with learnable queries, which serves as the understanding module to reason over multimodal conditions (video and text), 2) synchformer encoder (Iashin et al., 2024), which derives temporal latents to capture audio-visual temporal alignment, 3) flow-matching transformer with multiple MM-DiT blocks, which injects both semantic and temporal conditions into the generative process. and 4) separately-trained neural vocoder and VAE, to convert continuous audio latents into raw waveforms for high-fidelity audio synthesis.

## 4.2 MLLM SEMANTIC UNDERSTANDING

MLLMs are powerful reasoners with inherent strong reasoning and in-context learning capabilities to produce semantic information that guides generative models. Recent approaches (Liu et al., 2023; Chen et al., 2025a; Shi et al., 2024) demonstrate the effectiveness of MLLMs as backbones for perceiving and reasoning scenes and dynamic environments, while these pipelines introduce substantial complexity due to the multi-stage training to connect LLMs with diffusion backbones.

To avoid multi-stage and multi-task training, we freeze the MLLM backbone and include learnable queries to bridge understanding (i.e., MLLMs) and generative models (i.e., flow-matching models). Inspired by learnable prompts and queries (Pan et al., 2025; Gao et al., 2023), we prepend a set of learnable tokens to the input sequence, allowing the frozen MLLMs to incorporate newly adapted knowledge without fine-tuning. As illustrated in Figure 3(a), we randomly initialize and concat the learnable queries after the video and text tokens and query out the conditions for generation. For compatibility, we apply causal masking over the entire sequence instead of enabling full attention only for specific tokens. In this work, we use Qwen-2.5-VL-7B (Bai et al., 2025) as a backbone, which demonstrates strong video-text understanding capabilities in large-scale training.

## 4.3 GENERATIVE FLOW-MATCHING

Flow-matching generative models (Albergo & Vanden-Eijnden, 2022; Liu et al., 2022) are probabilistic models that fit the data distribution $p(x)$ by denoising in the data latent space. It encodes the original high-dim data $x$ into low-dim latent $z = \mathcal{E}(x)$, where the forward and reverse processes are performed in minimizing the trajectory curvature and connecting data and noise on a straight line.

As shown in Figure 3(c), we employ MM-DiT (Esser et al., 2024) as a triple-stream MMDiT architecture to showcase the multimodal joint training paradigm that jointly considers video, audio and text modalities within a unified transformer framework. Tokens from MLLM outputs and modality encoders (e.g., learnable queries and synchrony representations) are concatenated across modalities and interact via joint scaled dot-product attention. To encode temporal dynamics, we apply relative positional encoding via rotary positional embeddings (RoPE) (Su et al., 2024; Heo et al., 2024). For classifier-free guidance (CFG), we apply random conditional feature dropping to text or video conditions independently, enabling the model to jointly learn both conditional and unconditional objectives.

### 4.4 TEMPORAL ALIGNMENT PERFERENCE POST-TRAINING

Beyond semantic understanding, another major challenge for VT2A generation is achieving robust temporal alignment (Cheng et al., 2025; Luo et al., 2023) between input video and output audio. Although prior works (Wang et al., 2024) explore contrastive audio–visual objectives and high-frame-rate visual features to model temporal relations, audio–visual synchrony remains a persistent bottleneck: temporal cues in learned video representations are often weak, making synchrony difficult to capture reliably without explicit human priors or preference signals.

To strengthen temporal alignment, we perform preference post-training using synchrony feedback, aligning the generator with prior knowledge and preferences on visual–audio synchrony. We compare several alignment strategies, including supervised fine-tuning (SFT), reward-weighted regression (RWR) (Peng et al., 2019), and direct preference optimization (DPO) (Rafailov et al., 2023; Liu et al., 2025b). Models post-trained with preference optimization consistently improve temporal synchrony and outperform strong baselines. More details are provided in Section 6.1.

**Supervised fine-tuning (SFT)** selects the highest-reward (winner) sample in each group and optimizes toward desirable outputs, and the flow-matching regression loss is

$$\mathcal{L}_{\mathrm{SFT}}(\theta) = \mathbb{E}_{x_0, y, t}\left[ \| v^\star(x_t, y, t) - v_\theta(x_t, y, t)\|_2^2 \right],$$

where $v^\star(x_t, y, t)$ is the supervision velocity, and $v_\theta(x_t, y, t)$ is the predicted velocity field under parameters $\theta$.

**Reward-weighted regression (RWR)** reweights samples by a softmax over rewards within each group, thereby performing reward-weighted likelihood maximization. With reward-weighted regression, the loss becomes:

$$\mathcal{L}_{\mathrm{RWR}}(\theta) = \mathbb{E}_{x_0, y, t}\left[ \exp\bigl(r(x_0, y)\bigr) \| v^\star(x_t, y, t) - v_\theta(x_t, y, t)\|_2^2 \right],$$

where $r(x_0, y)$ is the reward associated with $(x_0, y)$. By weighting samples in this way, the flow-matching model emphasizes high-reward examples, analogous to reward-weighted likelihood maximization in flow-matching training.

**Direct performance optimization (DPO)** aligns diffusion models with pairwise preferences. For each condition $y$, we form a preference pair $x_0^w, x_0^l$ where $x_0^w$ (winner) is preferred to $x_0^l$ (loser) by a synchrony reward. Concretely, letting $x_t^w, x_t^l$ denote their noised states and $v_w^\star := v^\star(x_t^w, y, t)$, $v_l^\star := v^\star(x_t^l, y, t)$ the corresponding training targets, the loss contrasts the winner/loser errors relative to the reference model to amplify synchronized generations.

$$\mathcal{L}_{\mathrm{DPO}}(\theta) = -\mathbb{E}_{\{x_0^w, x_0^l, y\} \sim \mathcal{D}, \, t}\left[ \log \sigma\Bigl( -\frac{\beta_t}{2}\left(\Delta_t^w - \Delta_t^l\right)\Bigr)\right],$$

where $\Delta_t = \|v^\star(x_t, y, t) - v_\theta(x_t, y, t)\|_2^2 - \|v^\star(x_t, y, t) - v_{\mathrm{ref}}(x_t, y, t)\|_2^2$, and $\sigma(\cdot)$ denotes the logistic sigmoid, $x_t$ denotes the state at time $t \in [0, 1]$ in an flow-matching process, $v_{\mathrm{ref}}(\cdot, y, t)$ is a frozen reference model, and $v^\star(\cdot, y, t)$ is the FM supervision computed from $(x_0, t)$. $\beta$ is the hyperparameter to control the trade-off between the strength of the policy update and distance to the pretrained model.

## 5 TRAINING AND EVALUATION

### 5.1 DATASET

In training, we use VGGSound-Think as our primary video-text-to-audio (VT2A) training corpus. For audio-text data, we aggregate pairs from AudioSet-SL (Gemmeke et al., 2017), Freesound, and

AudioCaps (Kim et al., 2019), where the visual inputs are set to null tokens, resulting in a diverse corpus for training multimodal models. For evaluation (Luo et al., 2024; Iashin & Rahtu, 2021), we adopt the VGGSound and AudioCaps test set as the standard benchmark and further probe reasoning on the VGGSound-Think test split. To assess generalization, we report results on the Movie Gen Audio Bench dataset (Polyak et al., 2025) in Appendix I.

## 5.2 MODEL CONFIGURATIONS

For the MLLM understanding module, we build on the strong open-source Qwen2.5-VL-7B-Instruct backbone (Bai et al., 2025) and train a set of learnable queries $\mathcal{Q} \in \mathbb{R}^{N \times D}$, where we use $N = 77$ query tokens and $D$ equals the MLLM hidden dimension. For flow-matching models, the base learning rate is set to 0.005. To sample from the flow transformer, we use torchdiffeq (Chen et al., 2018) package to implement the ODE solvers with a step size of 0.04.

We train with a batch size of 512 for 200K optimization steps, followed by 100K steps of preference-based post-training. With this large batch size, the full training run converges in approximately 36 hours. In post-training, we apply LoRA (Hu et al., 2022) with rank 64 to the linear layers of the Transformer. Unless specified, we report results with our 160M-parameter post-trained model. Full hyperparameter settings are provided in Appendix B.

For synchronization feedback, we leverage an audio–visual temporal alignment classifier from Luo et al. (2024) as the reward model: given an audio–video pair $(x_0, y)$, the classifier predicts an alignment score ($\uparrow$), and we define the reward as $r(x_0, y) = \textbf{Align}(x_0, y)$. Using a different model/metric among post-training and evaluation reduces the risk of reward hacking or overfitting to the reward model's distribution.

## 5.3 EVALUATION METRICS

We evaluate models using objective and subjective metrics over audio quality, semantic alignment (text/video-audio), and audio-visual temporal synchrony. For fidelity, we report Frechet distance (VGG) ($\downarrow$), KL divergence ($\downarrow$), and Inception Score (PANNs) ($\uparrow$). For semantic alignment, we calculate 1) ImageBind (Girdhar et al., 2023) score (IB) ($\uparrow$) - measuring similarity between the input video and the audio; 2) CLAP (Elizalde et al., 2023) score ($\uparrow$) – measuring alignment between text and audio. For temporal alignment, the synchronization score (DeSync) ($\downarrow$) from Synchformer (Iashin et al., 2024) quantifies the misalignment between the audio and video.

For subjective metrics, we conduct crowdsourced human evaluations on Amazon Mechanical Turk, where raters are asked to rate MOS (mean opinion score) on a 20-100 Likert scale. We report MOS-Q (perceived audio quality) and MOS-F (perceived video–audio fit/synchrony), each with 95% confidence intervals (CI). Additional details are provided in Appendix G.

## 6 RESULTS

## 6.1 MAIN RESULTS

**Video-to-Audio Generation** For video–text-to-audio (VT2A) generation, we adopt the VGGSound test set as the standard benchmark and compare **ReasonAudio** with state-of-the-art systems, including Diff-Foley (Luo et al., 2024), Frieren (Wang et al., 2024), Im2Wav (Sheffer & Adi, 2023), MMAudio (Cheng et al., 2025), Tell What You Hear From What You See (Liu et al., 2024), Foley-Gen (Mei et al., 2024), and ThinkSound (Liu et al., 2025a). Table 2 summarizes the comparison and yields three observations:

1) **Semantic alignment.** ReasonAudio achieves strong cross-modal coherence, with an ImageBind score of $0.30$ (video–audio) and a CLAP score of $0.23$ (text–audio), indicating that the MLLM-backed understanding module helps generate audio well aligned with both modalities. 2) **Temporal alignment.** ReasonAudio attains state-of-the-art synchrony with DeSync $= 0.29$, demonstrating the effectiveness of preference optimization which aligns generative models with visual synchrony preferences. We discuss more details in Section 6.3. 3) **Audio quality.** ReasonAudio delivers perceptual quality comparable to strong baselines, achieving FD $= 1.89$ and KL $= 1.80$, suggesting that

| Model | Perceptual Quality | | | Semantic | | Temporal | Subjective Evaluation | |
|---|---|---|---|---|---|---|---|---|
| | FD (↓) | KL (↓) | IS (↑) | CLAP (↑) | IB (↑) | DeSync (↓) | MOS-Q (↑) | MOS-F (↑) |
| Diff-foley | 8.29 | 3.15 | 10.8 | 0.12 | 0.19 | 0.81 | 69.7±0.9 | 72.6±1.2 |
| Frieren | **1.34** | 2.53 | 12.3 | 0.19 | 0.22 | 0.89 | **82.3±1.3** | 76.2±0.8 |
| V2A-Mapper | 1.95 | 2.42 | 13.1 | 0.13 | 0.24 | 1.04 | 75.6±2.9 | 73.6±1.9 |
| MMAudio | 1.76 | **1.66** | 13.2 | 0.22 | **0.31** | 0.44 | 80.2±1.5 | 80.3±1.7 |
| ThinkSound | 2.43 | 2.46 | 12.5 | 0.19 | 0.26 | 0.63 | 77.5±2.1 | 79.4±1.5 |
| VATT-Gemma-T | 1.64 | 1.95 | 12.8 | 0.22 | 0.26 | 0.77 | - | - |
| Foleygen | 2.83 | 2.13 | 11.7 | 0.17 | 0.23 | 0.86 | - | - |
| ReasonAudio-Small | 1.89 | 1.80 | **16.9** | 0.23 | 0.30 | 0.29 | 80.5±1.3 | 83.3±1.6 |
| ReasonAudio-Large | 1.56 | 1.75 | 15.4 | **0.24** | **0.31** | **0.28** | 81.3±1.1 | **84.6±1.4** |

Table 2: Video-to-audio results on VGGSound testset. Following the common practice (Cheng et al., 2025), Diff-Foley, Im2Wav, and Frieren are conditioned on video, whereas MMAudio, ThinkSound, and ReasonAudio are conditioned on video and text. The best result is in bold and the second best result is underlined.

| Model | Perceptual Quality | | | Semantic | Subjective Evaluation | |
|---|---|---|---|---|---|---|
| | FD (↓) | KL (↓) | IS (↑) | CLAP (↑) | MOS-Q(↑) | MOS-F(↑) |
| Make-An-Audio 2 | **1.42** | 1.24 | 9.6 | 0.28 | **82.6±0.8** | 72.3±1.8 |
| Tango 2 | 2.96 | **1.16** | 10.2 | 0.32 | 75.9±1.6 | 83.1±1.1 |
| SoundCTM | 3.31 | 1.61 | 9.7 | 0.31 | 74.2±0.9 | 80.3±1.4 |
| MMAudio | 2.53 | 1.47 | 11.0 | 0.33 | 78.2±1.5 | 82.1±1.2 |
| ReasonAudio-Small | 2.36 | 1.43 | **11.7** | 0.34 | 79.6±1.1 | 84.6±0.9 |
| ReasonAudio-Large | 1.88 | 1.42 | 11.5 | **0.35** | 80.8±1.3 | **85.2±1.0** |

Table 3: Text-to-audio evaluation results on Audiocaps testset. The best result is in bold and the second best result is underlined.

improved semantic and temporal alignment does not come at the expense of audio fidelity. More detailed comparisons on ReasonAudio's video-to-audio generation have been attached in Appendix C.

**Text-to-Audio Generation** The video-text-to-audio framework is capable of text-to-audio synthesis without additional fine-tuning. We adopt the AudioCaps test set (Kim et al., 2019) as the standard benchmark and compare ReasonAudio with TANGO 2 (Majumder et al., 2024), Make-An-Audio 2 (Huang et al., 2023), SoundCTM (Saito et al., 2024), and MMAudio (Cheng et al., 2025) and present the comparison in Table 3. Consistent with our findings in video-to-audio generation, we make two key observations: 1) In terms of audio quality, Make-An-Audio 2 presents a slightly better FD of $1.42$. We hypothesize this gap stems from resampling: ReasonAudio generates 44 kHz audio that is downsampled to 16 kHz for VGG-based feature extraction, whereas the baseline natively outputs 16 kHz audio and thus avoids resampling-induced degradation. 2) On text-audio semantic alignment. ReasonAudio attains state-of-the-art performance with a CLAP score of $0.34$. This highlights the effectiveness of MLLMs as understanding components, which strengthen semantic reasoning and enable the model to generate faithful audio aligned with both text and video descriptions.

## 6.2 VIDEO-TO-AUDIO REASONING RESULTS

Text annotations in existing audio–visual benchmarks (e.g., VGGSound-test) often carry weak semantics and thus limit semantic reasoning. To provide an in-depth evaluation of semantic following in video-and-text-to-audio generation, we additionally evaluate on **VGGSound-Think** test set, which allows us to assess a model's ability to capture acoustic hint and understand descriptions on audio–visual relations. Further comparisons with Movie Gen Audio are included in the Appendix I.

**Quantitative results** We compare ReasonAudio with systems featuring advanced semantic understanding, including DeepSound-V1 (Liang et al., 2025) and ThinkSound (Liu et al., 2025a). As shown in Table 4, ReasonAudio achieves strong text–audio seman-

| Model | FD (↓) | CLAP (↑) | IB(↑) |
|---|---|---|---|
| DeepSound-V1 | 2.55 | 0.23 | 0.27 |
| ThinkSound | 2.67 | 0.25 | 0.29 |
| ReasonAudio | 2.09 | 0.28 | 0.32 |
| ThinkSound (Train) | 2.49 | 0.24 | 0.30 |

Table 4: Video-to-audio reasoning results on VGGSound-Think testset.

tic alignment with a CLAP score of 0.28 and robust video–audio coherence with an ImageBind score of 0.32, indicating that it generates audio well aligned with both text and video. We also train ThinkSound using the VGGSound-Think annotations, ensuring that the text conditioning is consistent with the ReasonAudio training procedure. As shown in Table, ReasonAudio achieves stronger text–audio semantic alignment (CLAP: 0.28) and more robust video–audio coherence (ImageBind: 0.32), suggesting the effectiveness of learnable prompts derived from strong MLLMs, which provide robust multimodal understanding and serve as an effective bridge between understanding and generation.

Unlike baselines that rely on multi-stage pipelines where MLLMs explicitly predict captions before connecting to diffusion models, our approach leverages learnable prompts, thereby avoiding the complexity of multi-stage training/inference and the need to tightly couple LLMs with diffusion backbones.

**Case study** Beyond quantitative evaluation, we provide case studies to qualitatively examine the model's ability to 1) reason over acoustic hints and 2) understand descriptions of audio–visual relations. We attach the full prompts in Appendix F, and we have the following findings:

- **Hint reasoning (Figure 1(a)).** Our model (bottom) successfully interprets the acoustic hint *"someone calls 911 and stays here until help arrives"* and generates the siren sounds without the sound source (i.e., police siren) being explicitly specified. In contrast, the baseline fails to connect the text to the expected emergency context. The result highlights stronger semantic understanding empowerd by MLLMs: it maps the implicit "911" cue to an appropriate acoustic without the object (siren) being explicitly named.

- **Understanding descriptions on audio–visual relations (Figure 1(b)).** The baseline (top) primarily reproduces broadband snow/wind energy while ignoring the off-screen sound instructions *"human speech"*. In contrast, our model introduces salient off-screen human elements: the spectrogram shows intermittent narrowband ridges and energy peaks aligned with human vocal bursts, together with sustained turbulence consistent with wind. These results highlight the tri-modal audio–visual–text alignment, where the model better matches the textual specification of multi-object (snow/wind/human) and on/off-screen relationships.

### 6.3 ANALYSIS AND ABLATION STUDIES

To verify the effectiveness of several designs in ReasonAudio, including MLLMs, reward post-training and scalability, we conduct ablation studies and discuss the key findings as follows.

**Scalability.** We report results for two model sizes: 160M (M) and 750M (L) parameters. As shown in Table 2 and Table 3, scaling up improves most metrics. For example, in the video-to-audio setting, increasing the model from 160M to 750M yields a clear improvement in perceptual quality (FD from 1.89 to 1.56), while the gain in audio–visual alignment is minor (DeSync from 0.43 to 0.42). This suggests that larger capacity primarily boosts fidelity, with limited gains for temporal synchrony.

**MLLMs For Understanding.** We ablate the model's understanding module in VGGSound-Think testing set by replacing the MLLM with (i) CLIP features, (ii) a text-only LLM (same MLLM with the visual stream removed), (iii) a hybrid of CLIP and text-only LLM, and (iv) Qwen-Omni-7B (Xu et al., 2025) as MLLM backbone. As can be seen in Table 5, the CLIP-based variant yields lower semantic scores (CLAP and IB), suggesting that contrastive features hinder reasoning over dynamic, long-range

| Model | FD ($\downarrow$) | CLAP ($\uparrow$) | IB ($\uparrow$) |
|---|---|---|---|
| ReasonAudio | 2.09 | 0.28 | 0.32 |
| CLIP | 1.92 | 0.24 | 0.25 |
| LLM | 2.11 | 0.27 | — |
| CLIP+LLM | 2.06 | 0.28 | 0.28 |
| ReasonAudio-Omni | 1.98 | 0.29 | 0.31 |

Table 5: Ablations of understanding components.

context. The text-only LLM also limits the performance as it reasons solely over text, making it unable to capture audio-visual relations. The hybrid baseline achieves comparable text–audio alignment (CLAP), indicating that an LLM can effectively reason over dynamic, long-range context comparable to MLLMs. However, it yields lower audio–visual semantic alignment (IB), suggesting

that replacing an MLLM with a CLIP-based visual encoder weakens cross-modal understanding. Omni ReasonAudio presents improvement in fidelity (FD) and text-audio alignment (CLAP), while witnessing a degradation in video alignment (IB). Omni MLLM is trained with video–audio supervision and is therefore stronger at tri-modal (text–video–audio) reasoning and alignment.

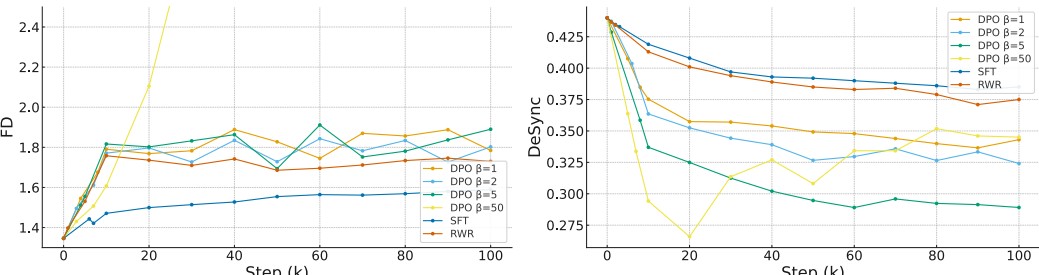

Figure 4: Reward post-training. As discussed in Section 5.2, we initialize from the flow-matching pre-trained model and optimize the reward objective for 100K steps.

**Reward Post-Training.** We compare preference post-training among DPO, SFT, and RWR strategies. As can be seen in Figure 4, the DPO models demonstrate the distinct temporal alignment improvement and DPO with $\beta = 5$ yields the best temporal alignment scores, indicating that DPO fine-tuning is more effective than SFT and RWR in leveraging preference feedback. We also note that preference post-training introduces a slight degradation in audio quality measured by FD. Unlike SFT, which only optimizes toward desirable (winner) outputs, both DPO and RWR also incorporate undesirable (loser) outputs during training, enabling more effective preference learning and resulting in superior alignment performance.

For DPO fine-tuning, $\beta$ controls the trade-off between the strength of the policy update and fidelity to the pretrained model. We ablate $\beta \in \{1, 2, 5, 50\}$. Increasing $\beta$ accelerates reward improvement, but beyond $\beta = 5$ we observe a noticeable drop in audio quality (higher FD), indicating overfitting toward the reward model. Although the alignment–fidelity trade-off is observed, 1) ReasonAudio-Large ranks the second-best performance model in audio quality (FD), and 2) ReasonAudio-Small and ReasonAudio-Large attain state-of-the-art performance with an IS score of 11.7, 11.5. To balance adaptively during post-training, one alternative way is to dynamically calibrate $\beta$ at data quality considerations (Wu et al., 2024), where $\beta$ is adaptively decreased for closely-matched pairwise data (i.e., low gap data) to facilitate assertive updates, and increased for easily-discriminated pairs (i.e., high-gap data)

## 7 CONCLUSION

In this work, we proposed ReasonAudio, an MLLM-empowered flow-matching generative model with stronger semantic understanding and robust temporal alignment. We addressed the scarcity of semantically rich annotations by constructing VGGSound-Think, a tri-modal (video–text–audio) dataset enriched with acoustic hints and descriptive annotations of audio–visual relations (objects, on-/off-screen attribution). Empowered by MLLMs with learnable queries, ReasonAudio excelled in understanding multimodal conditions (video and text) and following complex semantics. Furthermore, preference optimization (Flow-DPO, Flow-RWR) effectively aligned generative models with visual synchrony preferences, leading to enhanced audio–visual temporal alignment. Extensive experiments demonstrate state-of-the-art VT2A performance, with substantial gains in both semantic alignment and temporal synchronization.

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

# Appendices

## ReasonAudio: Semantic Reasoning and Temporal Synchrony in Video–Text-to-Audio Generation

## A  DATA SETUP

We present the statistics for multimodal datasets as follows:

| Conditions | Dataset | Hours |
|---|---|---|
| Video-Text | VGGSound-Think | 450 |
| Text | Audioset (Gemmeke et al., 2017) | 262 |
| Text | Freesound | $\sim 1200$ |
| Text | AudioCaps (Kim et al., 2019) | 128 |

Table 6: Statistics for datasets used for training.

## B  MODEL CONFIGURATIONS

We list the model hyper-parameters of ReasonAudio in Table 7.

| Hyperparameter | | ReasonAudio |
|---|---|---|
| M | Transformer Layer | 12 |
| | Transformer Embed Dim | 448 |
| | Transformer Attention Headers | 7 |
| | Number of Parameters | 160 M |
| L | Transformer Layer | 21 |
| | Transformer Embed Dim | 896 |
| | Transformer Attention Headers | 14 |
| | Number of Parameters | 750 M |
| BigVGAN Vocoder | Upsample Rates | [5, 4, 2, 2, 2, 2] |
| | Hop Size | 320 |
| | Upsample Kernel Sizes | [9, 8, 4, 4, 4, 4] |
| | Number of Parameters | 121.6M |

Table 7: Hyperparameters of ReasonAudio.

### B.1  VAE

The audio encoder $E$ takes mel-spectrogram $x_a$ as input and outputs compressed latent $z = E(x_a)$. The audio decoder $D$ reconstructs the mel-spectrogram signals $\tilde{x_a} = D(z)$ from the compressed representation $z$. Different from other modalities, we use an audio VAE with 1D-convolution to improve the model's capacity for variable-length audio. VAE solves the problem of excessive smoothing in mel-spectrogram reconstruction through adversarial training with a discriminator.

The training objective is to minimize the weighted sum of reconstruction loss, GAN loss, and KL penalty loss. To this end, ReasonAudio takes advantage of the VAE to predict self-supervised representations instead of waveforms. It largely alleviates the challenges of modeling long continuous data and guarantees high-level semantic understanding.

## B.2 VOCODER

We train a BigVGAN (Lee et al., 2022) vocoder from scratch for the spectrogram to waveform generation. The synthesizer includes the generator and multi-resolution discriminator (MRD). The generator is built from a set of look-up tables (LUT) that embed the discrete representation and a series of blocks composed of transposed convolution and a residual block with dilated layers. The transposed convolutions upsample the encoded representation to match the input sample rate.

## C  VIDEO-TO-AUDIO (V2A) GENERATION

For video-to-audio (V2A) generation without text, we use the VGGSound test set as the standard benchmark and evaluate ReasonAudio-Small in terms of perceptual quality, video–audio semantic alignment, and audio–visual temporal synchrony. As shown in Table, ReasonAudio under pure V2A conditioning achieves strong cross-modal coherence, with an ImageBind score of $0.24$ and a DeSync score of $0.29$, and maintains comparable perceptual quality (FD) to competitive baselines.

In summary, while our structural prompts explicitly provide acoustic hints and structured audio–visual relation descriptions (e.g., object grounding and on-/off-screen attribution), the model remains robust even without text guidance for two reasons: 1) instead of directly relying on textual representations, ReasonAudio conditions on learnable prompts derived from strong MLLMs, which provides robust multimodal understanding and serves as an effective bridge between understanding and generation; 2) Besides, we apply random conditional feature dropping during classifier-free guidance (CFG) training, which improves cross-modal generalization while preserving the flexibility of using an text input.

| Model | FD ($\downarrow$) | KL ($\downarrow$) | IS ($\uparrow$) | IB ($\uparrow$) | DeSync ($\downarrow$) |
|---|---|---|---|---|---|
| ReasonAudio-VT2A | 1.89 | 1.80 | 16.9 | 0.23 | 0.29 |
| ReasonAudio-V2A | 1.91 | 1.83 | 16.7 | 0.24 | 0.29 |

## D  VGGSOUND-THINK DATASET VALIDATION

### D.1 ANNOTATION COLLETCTION

Each sample is annotated through a structured, step-by-step procedure:

- **Coarse-grained acoustic gist.** For each audio clip, we prompt GPT-4o with:

  You are given a video (frames) and its audio. Write a *single-sentence* coarse acoustic gist that summarizes the dominant sound events and overall ambience *without naming specific sound-producing objects* (e.g., avoid "car", "dog", "siren"). Instead, describe sound *attributes* such as pitch, timbre, rhythm, intensity, continuity, and background/foreground. Output only the one sentence.

  to obtain a high-level hint that captures the main acoustic content while remaining object-agnostic.

- **Fine-grained sound grounding.** We prompt GPT-4o with:

  You are given a video (frames) and its audio. Write **one concise paragraph** (2–4 sentences) that **grounds salient sounds to visible entities** and describes how the sound evolves over time.
  **Requirements:** (1) Mention only entities that are clearly supported by the video (do NOT invent objects). (2) Explicitly connect each salient sound to its most likely on-screen source; if the source is likely off-screen, say so. (3) Include temporal progression using natural phrasing (e.g., "begins", "rises", "fades", "as X happens"). (4) Prefer concrete audio descriptors (e.g., "wailing siren", "engine chugging", "faint speech", "reverberant") and visual evidence cues (e.g., "a truck is shown", "mouth movement", "vehicle starts moving").

  to align sound events with specific visible objects, refining the gist into grounded descriptions.

- **Structured audio–visual relation annotations.** For each grounded object, we prompt GPT-4o with:

> You are given a video (frames) and its audio. Produce a **structured, line-based annotation** of the main sound-producing entities.
> **Step 1: Identify objects.** List 2–6 entities that plausibly produce salient sounds. Name each entity with a short **snake_case** identifier (e.g., `mechanical_siren`, `human_speech`, `fire_truck_engine`). Use only entities supported by the video.
> **Step 2: For each object, output exactly ONE line in the following format:** `<Object '{object_id}' screen={on_screen/off_screen}> {audio_keywords}`
> **audio_keywords requirements:** - Provide 3–8 terse keywords/phrases separated by spaces. - Focus on sound attributes and events, e.g., `wailing, rise_fall, reverberant, faint, rumble, chugging, idling`. - If helpful, include a single interaction cue such as `masked_by_X` or `overlaps_with_X` (where X is another `object_id`).

to generate descriptive keywords and explicit audio–visual relations, including on-/off-screen attribution and interactions among objects.

• **Filtering and validation.** We conduct a strict manual check where 5% of the full dataset is randomly sampled for human review as a quality control measure. This procedure serves as a check on our filtering rules and helps ensure the dataset's overall reliability. During inspection, reviewers evaluate both technical alignment (e.g., audio–visual synchronization and consistency) and perceptual correspondence between the sound and the visual content.

## D.2 GPT-4O COMPARISON

We evaluate and compare with VideoLLaMA3 (Zhang et al., 2025), an open-source MLLM, to extract temporal and semantic information from videos. Using the same example in Figure 2 ("fire truck siren"), we provide a qualitative comparison in the table.

| | GPT-4o | VideoLLaMA3 |
|---|---|---|
| **Fine-grained sound grounding** | The scene is an urban street or parking lot at dusk, featuring a vintage red fire truck. A mechanical siren begins, winding up and then rapidly down in pitch. As the siren fades, a faint voice and the low, chugging rumble of the truck's engine becomes prominent as the vehicle slowly begins to drive away. | A fire truck drives on a city street. The siren is loud and varies in pitch, with engine/road noise underneath. After it passes, the siren weakens and background/engine noise remains. |
| **Structured audio–visual relations** | `<Object 'mechanical_siren' screen=on_screen>` wailing rise_fall; reverberant `<Object 'human_speech' screen=on_screen>` faint; brief; reverberant `<Object 'antique_fire_truck_engine' screen=on_screen>` idling rumble; chugging | fire truck (red), flashing lights siren: loud, up/down pitch, then quieter engine/road noise present |

VideoLLaMA3 demonstrates strong high-level multimodal understanding and conversational ability. However, it is less efficient at following our instruction format: (1) for fine-grained sound grounding, it tends to produce shorter and less precise descriptions; and (2) for structured audio–visual relation annotations, its outputs contain fewer explicit fields and are less consistently structured.

## D.3 DIFFERENCES IN TEXT FORMAT BETWEEN VGGSOUND-THINK AND VGGSOUND

The VGGSound-Think reasoning captions are written in natural language and comprise three components: (1) a coarse-grained acoustic gist, (2) fine-grained sound grounding, and (3) structured audio–visual relation annotations. The key format differences relative to standard VGGSound cap-

tions mainly lie in the additional acoustic hints and the explicit structured audio–visual relations (e.g., object grounding and on-/off-screen attribution).

To test the text impacts on generation, we gather results from Table 1 and Table 2, respectively generated in the VGGSound test and VGGSound-Think test set, to assess a model's ability to capture acoustic hints and understand structured audio–visual relation descriptions. We evaluate ReasonAudio-Small in terms of perceptual quality (FD, KL) and video–audio semantic alignment (IB). Since the ground-truth texts differ across the two settings, the CLAP is less informative and thus omitted. As shown in the Table, ReasonAudio (VGGSound-Think) achieves stronger video–audio semantic alignment and enhanced perceptual quality, showcasing the benefits of learning semantically rich textual descriptions and the generalization to different text descriptions.

| Model | FD ($\downarrow$) | KL ($\downarrow$) | IB($\uparrow$) |
|---|---|---|---|
| ReasonAudio (VGGSound-Think) | 2.09 | 1.38 | 0.32 |
| ReasonAudio (VGGSound) | 2.36 | 1.43 | 0.31 |

Table 8: Ablations on text format.

ReasonAudio generalize well to different text formats (structured or plain text) for two main reasons: 1) instead of directly relying on textual representations, ReasonAudio conditions on learnable prompts derived from strong MLLMs, where the MLLMs provides robust multimodal understanding across various text formats; 2) Besides, ReasonAudio is jointly trained on both structured texts (VT2A) and plain texts (T2A) data, which improves cross-modal generalization and preserves the flexibility of text format.

## E   ABLATION STUDIES ON THE NUMBER OF QUERIES

For the MLLM understanding module, we use $N = 77$ learnable queries intended to enable a fair comparison between LLM and CLIP-based conditioning by matching the representation shape $Q \in \mathbb{R}^{N \times D}$, where we use $D$ equals the MLLM hidden dimension.

We also ablate the understanding module on the VGGSound-Think test set by varying the number of learnable queries. As can be seen in Table, reducing the number of queries consistently degrades semantic scores (CLAP and IB). Increasing $N$ accelerates improvement, but gains saturate at $N = 128$, where we observe only marginal improvements, indicating that learnable queries effectively compress conditioning information into a fixed-length token set, providing both compact and semantically rich latent embeddings.

| Queries | FD ($\downarrow$) | CLAP ($\uparrow$) | IB ($\uparrow$) |
|---|---|---|---|
| 128 | 2.04 | 0.29 | 0.33 |
| 77 | 2.09 | 0.28 | 0.32 |
| 64 | 2.17 | 0.26 | 0.31 |
| 32 | 2.32 | 0.24 | 0.29 |

Table 9: Evaluation varying the number of learnable queries

## F   VGGSOUND-THINK EXAMPLES

Here we provide the prompt examples of VGGSound-Think, showcasing the strong capabilities to reason over acoustic hints:

*<HINT>Someone calls 911 and stays here until help arrives.</HINT> <TRACK name='caller_waiting' screen=on_screen> <KEYWORDS> quiet room, anxious pacing, soft rustle, handset handling </KEYWORDS> </TRACK> <TRACK name='approaching_emergency_vehicle' screen=off_screen> <KEYWORDS> cycling tonal pattern, rising loudness, approach-pass, outdoor </KEYWORDS> </TRACK> <TRACK name='dispatch_handset_exchange' screen=off_screen> <KEYWORDS> clipped*

*phrases, short bursts, handheld device, intermittent </KEYWORDS> </TRACK> <TRACK name='intersection_yield_dynamics' screen=off_screen> <KEYWORDS> brake rub, tire scrub, indicator ticks, staggered movement </KEYWORDS> </TRACK>*

*<HINT>Crisp, granular crunching under pressure is followed by a series of brief, energetic, high-pitched tonal bursts and indistinct murmurs. A powerful and sustained rush of air builds, suggesting rapid movement through an open, cold environment. </HINT> This video captures a first-person perspective of someone standing on and then tumbling down a snowy mountain slope. The audio begins with faint, distant shouts and wind, which are abruptly replaced by a loud, chaotic scraping and rushing sound as the person slides down the hill. This intense sound of friction and turbulence continues until the person comes to a stop, followed by a close-up, heavy exhale. The overall acoustics are dominated by the near-field, high-energy turbulent noise of the slide, contrasting with the initially distant environmental sounds. <TRACK name='wind' screen=off_screen> <KEYWORDS> broadband noise turbulent rushing </KEYWORDS> </TRACK> <TRACK name='distant_human_speech' screen=off_screen> <KEYWORDS> distant muffled indistinct chatter </KEYWORDS> </TRACK> <TRACK name='human_vocalization' screen=off_screen> <KEYWORDS> close_mic effort grunt breathy exhale </KEYWORDS> </TRACK> <TRACK name='sliding_on_snow' screen=on_screen> <KEYWORDS> friction scraping turbulent rush granular </KEYWORDS> </TRACK>*

## G  EVALUATION

To probe audio quality, we conduct the MOS-Q (mean opinion score) tests and explicitly instruct the raters to *"focus on examining the audio quality and naturalness."*. The testers present and rate the samples, and each tester is asked to evaluate the subjective naturalness on a 20-100 Likert scale.

To probe video-audio alignment, human raters are shown an audio and a video and asked *"Does the audio align with video faithfully?"*. They must respond with "completely", "mostly", or "somewhat" on a 20-100 Likert scale to score MOS-F.

Our subjective evaluation tests are crowd-sourced and conducted via Amazon Mechanical Turk. These ratings are obtained independently for model samples and reference audio. The screenshots of instructions for testers have been shown in Figure. We paid $8 to participants hourly and totally spent about $600 on participant compensation. A small subset of audio samples used in the test is available at `https://ReasonAudio.github.io/`.

Ratings are collected independently for model-generated samples and reference audio, and we recruit 20 raters with normal hearing. All samples (50 video-audio pairs per subject score) are presented in randomized order to mitigate ordering effects. We report each subjective metric as mean $\pm$ standard deviation (SD) in the main paper to reduce randomness, where SD reflects the variability of ratings across samples and raters.

## H  MOVIEGEN AUDIO GENERALIZATION

To assess generalization, we include additional qualitative visualizations of video-to-audio generation on the MovieGen Audio benchmark and report objective metrics (IS, IB, CLAP, and DeSync score) to quantify fidelity and alignment.

| Method | IS ↑ | IB ↑ | CLAP ↑ | DeSync ↓ |
|--------|------|------|--------|----------|
| MMAudio | 8.40 | 27.0 | 0.43 | 0.77 |
| ThinkSound | 8.64 | 29.6 | 0.45 | 0.76 |
| ReasonAudio | **8.96** | **32.8** | **0.46** | **0.59** |

Table 10: Automatic metrics for MovieGen Audio Generalization.

Compared to MMAudio and ThinkSound as baselines, ReasonAudio achieves strong text–audio semantic alignment with a CLAP score of 0.46 and robust video–audio coherence with an ImageBind score of 32.8, indicating better generalization to the MovieGen Audio benchmark and improved

adherence to semantic conditioning. The results highlight the advantage of using MLLMs as an understanding module directly bridge multimodal understanding and audio generation, strengthening end-to-end multimodal reasoning.

## I  MORE VISUALIZATION

In this section, we put more visualizations of video-to-audio generation results.

## J  LIMITATIONS

ReasonAudio adopts flow generative models for high-quality synthesis, and thus, multiple ODE refinements are required for better results. Besides, MLLMs typically require more GPU memory in training and inference. One of our future directions is to develop a lightweight and fast MLLM empowered flow-based transformer for accelerating sampling.

## K  USE OF LARGE LANGUAGE MODELS (LLMs)

In this work, MLLMs have the following usage:

- MLLMs are powerful reasoners with inherent strong reasoning and in-context learning capabilities to produce semantic information that guides generative models. We use Qwen-2.5-VL-8B (Bai et al., 2025) as backbone which demonstrates strong video-text understanding capabilities for perceiving and reasoning scenes and dynamic environments.
- To construct VGGSound-Think, we generate audio descriptions using GPT-4o (Hurst et al., 2024) which excells in multimodal understanding and conversations. Each sample is annotated through a structured, step-by-step procedure.

## L  REPRODUCIBILITY STATEMENT

We will release our code in the future. The ReasonAudio model that we build upon is publicly available through the SiT repository (Ma et al., 2024). To aid reproducibility, we have included an overview of the hyperparameters in Table 7.

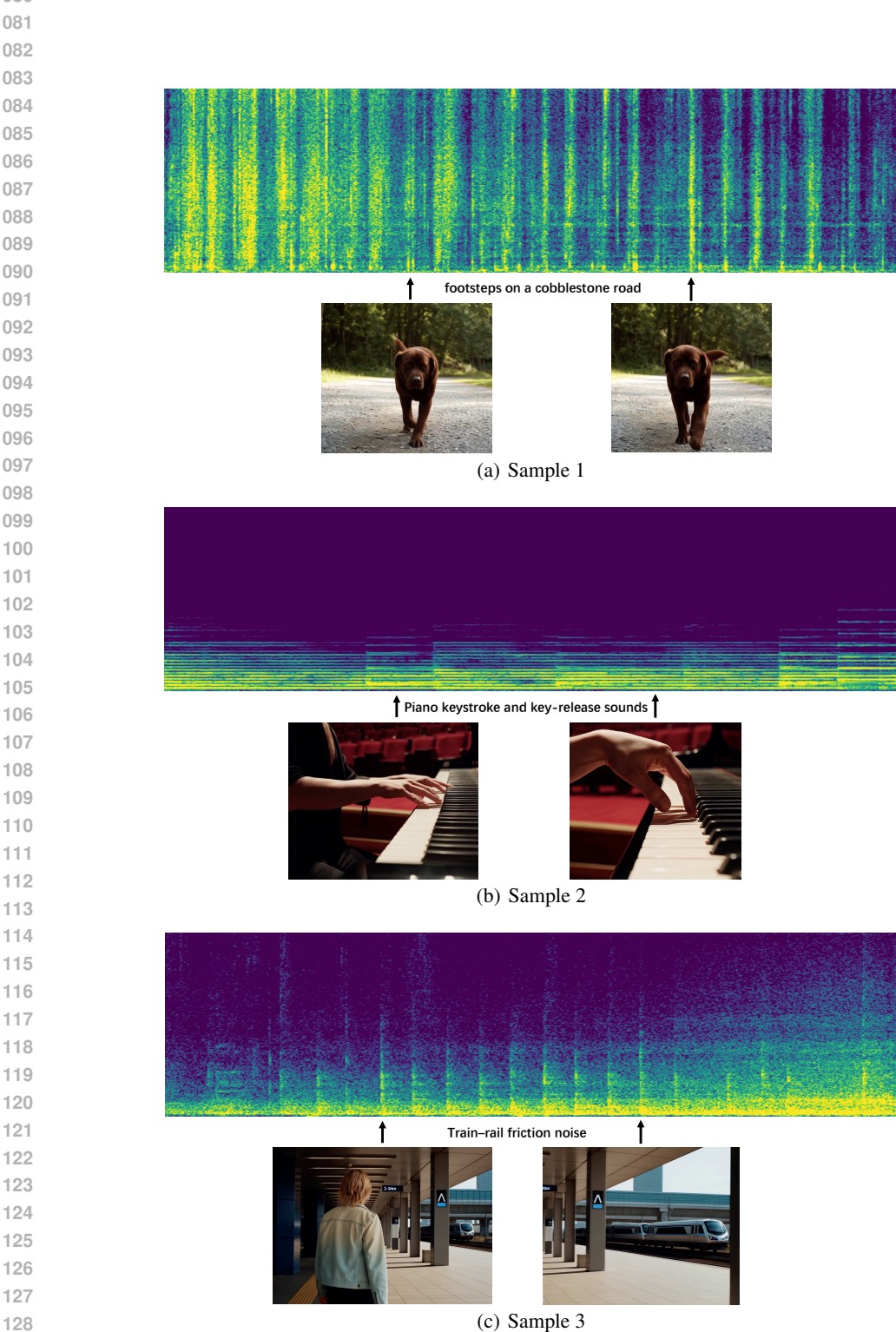

(a) Sample 1

(b) Sample 2

(c) Sample 3

Figure 5: Visualizations of video-to-audio generation in MovieGen Audio Bench.

