# OpenReview forum: "ReasonAudio: Semantic Reasoning and Temporal Synchrony in Video–Text-to-Audio Generation"
_ICLR.cc/2026/Conference — ICLR 2026 Conference Withdrawn Submission_

### Official Review · Reviewer_3J1P · 2025-10-26

**Soundness:** 3
**Presentation:** 2
**Contribution:** 3
**Rating:** 4
**Confidence:** 5

**Summary:**

This paper presents a new MLLM-empowered framework for video-text-to-audio (VT2A) generation, ReasonAudio. In addition, preference optimization (Flow-DPO or Flow-RWR) is adopted for enhancing temporal alignment between the video and audio modalities. The paper also introduces a dataset enriched with acoustic hints and audio-visual relation descriptions, VGGSound-Think, to address the scarcity of semantically rich tri-modal (video–text–audio) annotations in previous datasets. Experiment results show that ReasonAudio outperforms existing models in terms of semantic alignment and temporal synchronization in the VT2A generation task.

**Strengths:**

1. Although there is room for improvement in presentation, the paper itself is written well enough to make readers understand the proposed method and the experimental results.
2. The introduced methods sound reasonable.
3. The experiments are comprehensive, and the quantitative results are good.

**Weaknesses:**

1. There is a problem regarding the generated samples on https://ReasonAudio.github.io .
    - [1-a] Some samples generated by ReasonAudio cannot be played (as of October 25th, 2025).
    - [1-b] I felt the quality of the other generated samples was not as good as the quantitative result.
2. There are some unclear points for me.
    - [2-a] How were synchrony preferences obtained?
    - [2-b] L.256 (Section 4.3) says, "we randomly drop conditional features." Did you drop all conditional features simultaneously? Or, did you drop each conditional feature independently, as in MMAudio and ThinkSound?
    - [2-c] How was the difference in text format between VGGSound-Think and VGGSound handled? In the VGGSound-Think dataset, which is used for training a model, text descriptions are detailed and have a structure. On the other hand, text descriptions in the VGGSound dataset, which is used for evaluations, are simple and plain. I wonder how the proposed framework deals with this difference.

**Questions:**

- I would appreciate it if the authors could address Weaknesses 2-a, 2-b, and 2-c that I provided above.
- Additional questions about RoPE.
  - [3-a] Which type of RoPE is used for noisy audio latent and temporal latent, aligned RoPE (which [Mei et al. (2024)](https://openreview.net/forum?id=HZq8Gakf6e), MMAudio, and ThinkSound adopt) or non-aligned RoPE?
  - [3-b] Could you elaborate on why RoPE is applied to semantic queries within MMDiT? To my knowledge, one of the most popular MMDiT implementations is not to apply RoPE to text query tokens. [FLUX](https://github.com/black-forest-labs/flux) and MMAudio adopt this implementation. Did you decide the design empirically?

If the authors' response is convincing, I will raise my rating.

---

> ### Author Response · Authors · 2025-11-27
>
> ## **Response to Reviewer 3J1P (1/2)**
>
> We thank the reviewer for the constructive feedback and for considering our work as "paper is written well enough", "methods sound reasonable", and "experiments are comprehensive, and results are good". We understand that your concerns are mainly related to the paper's clarity, analysis, and ablation studies, and we hope our response resolves your concerns fully.
>
>
>
> **[About the generated samples]**
>
> Thanks for pointing out the display issue. We found that some filenames started with the character "_", and after a recent website update, files with this prefix could not be played correctly. We have fixed the issue and would be grateful if the reviewer could get a sense of it.
>
>
> **[About the synchrony preference]**
>
> Thanks for the reviewer's feedback that requests more explanation on the usage of synchronization feedback. We leverage an audio–visual temporal alignment classifier [1] as the reward model: given an audio–video pair $(x_0, y)$, the classifier predicts an alignment score ($\uparrow$), and we define the reward as $r(x_0,y)=\textbf{Align}(x_0,y)$. For evaluation, we report the synchronization score (DeSync) ($\downarrow$) from Synchformer, which measures audio–video misalignment and is distinct from our reward signal. Using a different model/metric for evaluation reduces the risk of reward hacking or overfitting to the reward model’s distribution.
>
> To train the classifier, [1] uses three types of pairs: 50\% of the pairs are real audio-visual pairs (true pair) labeled as 1, 25\% are audio-visual pairs from the same video but temporally shifted (temporal shift pair) labeled as 0, and the last 25\% are audio-visual pairs from different videos (wrong pair) labeled as 0. The resulting alignment classifier achieves 90\% test accuracy, indicating that it provides a meaningful signal for audio–visual synchronization, and it has also been used for evaluation in related work such as VATT-Gemma-T [2].
>
> [1] Diff-Foley: Synchronized Video-to-Audio Synthesis with Latent Diffusion Models.
>
> [2] Tell What You Hear From What You See - Video to Audio Generation Through Text.
>
>
> **[About the random drop in CFG]**
>
> Thanks for the reviewer's feedback that requests more explanation. We apply random conditional feature dropping to text or video conditions independently (similar to MMAudio and ThinkShound), so the model jointly learns conditional and unconditional objectives.

---

> ### Author Response · Authors · 2025-11-27
>
> ## **Response to Reviewer 3J1P (2/2)**
>
> **[About the differences in text format between VGGSound-Think and VGGSound]**
>
> Thanks for the reviewer's feedback that requests more explanation on the text format differences. The VGGSound-Think reasoning captions are written in natural language and comprise three components: (1) a coarse-grained acoustic gist, (2) fine-grained sound grounding, and (3) structured audio–visual relation annotations. The key format differences relative to standard VGGSound captions mainly lie in the additional acoustic hints and the explicit structured audio–visual relations (e.g., object grounding and on-/off-screen attribution).
>
> To test the text impacts on generation, we gather results from Table 1 and Table 2, respectively generated in the VGGSound test and VGGSound-Think test set, to assess a model's ability to capture acoustic hints and understand structured audio–visual relation descriptions. We evaluate ReasonAudio-Small in terms of perceptual quality (FD, KL) and video–audio semantic alignment (IB). Since the ground-truth texts differ across the two settings, the CLAP is less informative and thus omitted. As shown in the Table, ReasonAudio (VGGSound-Think) achieves stronger video–audio semantic alignment and enhanced perceptual quality, showcasing the benefits of learning semantically rich textual descriptions and the generalization to different text descriptions.
>
> | Model                      | FD (↓) | KL (↓) | IB (↑) |
> |---------------------------|-------:|-------:|-------:|
> | ReasonAudio (VGGSound-Think) | 2.09  | 1.38   | 0.32   |
> | ReasonAudio (VGGSound)       | 2.36  | 1.43   | 0.31   |
>
> ReasonAudio generalize well to different text formats (structured or plain text) for two main reasons:
> - 1) Instead of directly relying on textual representations, ReasonAudio conditions on learnable prompts derived from strong MLLMs, where the MLLMs provides robust multimodal understanding across various text formats;
> - 2) Besides, ReasonAudio is jointly trained on both structured texts (VT2A) and plain texts (T2A) data, which improves cross-modal generalization and preserves the flexibility of text format.
>
> **[About the RoPE]**
>
> Thanks for the reviewer’s feedback requesting clarification on our use of RoPE. We use positional embeddings to provide the attention layers with temporal information. As shown in Figure 3(c) in the main paper, we apply temporally aligned RoPE to the queries and keys of both the visual and audio streams (similar to MMAudio and ThinkSound).
>
> We include RoPE on the semantic queries based on empirical validation. Unlike MMAudio, where text conditioning is treated as temporally agnostic, ReasonAudio's semantic queries are produced from MLLM learnable-prompt outputs and represent fused video–text features. Since these features already carry temporal structure inherited from the video stream, we apply RoPE to the semantic queries (similar to the RoPE on video features in MMAudio).
>
> **Again, we appreciate the reviewer's valuable reviews and believe your concerns are mainly due to the paper's clearness, analysis, and ablation studies. In this rebuttal phase, we will try to address them carefully, and we would be grateful if you could update your score if our responses resolve your concerns.**

---

> > ### Comment · Reviewer_3J1P · 2025-11-27
> >
> > Thank you for your work. I went through all the reviewers' comments and your responses. I appreciate all of them.
> >
> > In ICLR, authors are allowed to update their manuscripts, including their supplementary material. You can increase the amount of the main body up to 10 pages (see [Author Guide](https://iclr.cc/Conferences/2026/AuthorGuide)). I recommend updating your manuscript by incorporating the additional explanations and experimental results you have prepared in response to the review comments.

---

> > > ### Comment · Reviewer_3J1P · 2025-11-28
> > >
> > > Thank you for revising the manuscript. I am considering raising my rating, but I would like to request an additional assessment.
> > >
> > > This issue is not originally from my review comment, but in the MovieGen Audio benchmarking (Appendix H), ThinkSound should be compared. The difference between ReasonAudio and ThinkSound has already been clarified, but many readers would be interested in comprehensive comparisons with ThinkSound.

---

> > > > ### Author Response · Authors · 2025-11-28
> > > > **Thanks for your positive feedback and considering raising scores!**
> > > >
> > > > Thank you again for your great efforts and valuable comments. We greatly thank the reviewer for providing constructive feedback. We will continue to improve our presentation based on reviewers' comments and suggestions. We have attached the comparison with ThinkSound in Appendix H. Compared to MMAudio and ThinkSound as baselines, ReasonAudio achieves strong text–audio semantic alignment with a CLAP score of $0.46$ and robust video–audio coherence with an ImageBind score of $32.8$, showing generalization to the MovieGen Audio benchmark and improved adherence to semantic conditioning. These results highlight the advantage of using MLLMs with learnable queries as the understanding module to directly bridge multimodal understanding and audio generation, strengthening end-to-end multimodal reasoning.
> > > >
> > > > Again, we greatly appreciate that you could raise your score, and believe that your valuable comments have improved the paper with detailed explanations and more precise presentations. We greatly appreciate the reviewer's time, efforts, and patience during this peer review stage.

---

### Official Review · Reviewer_D3zi · 2025-10-29

**Soundness:** 2
**Presentation:** 3
**Contribution:** 2
**Rating:** 4
**Confidence:** 5

**Summary:**

The paper proposes a VT2A framework that (i) augments VGGSound with semantically richer, structured captions (“VGGSound-Think”: hints + AV relations), (ii) uses a frozen MLLM with learnable queries to encode video+text conditions, and (iii) applies preference-based post-training (Flow-DPO/Flow-RWR) using a synchronization feedback signal to improve temporal A/V alignment. On VGGSound and AudioCaps, it shows strong semantic alignment (CLAP/IB), competitive fidelity (FD/KL/IS), better DeSync, and improved MOS-Fit.

**Strengths:**

1. Semantic reasoning + audio-visual temporal alignment are the two real pinpoints in V+T->A, the proposed method is clear and targets at addressing both problems.

2. Applying DPO/RWR to a flow model for temporal alignment is interesting; the β ablation and SFT/RWR/DPO comparison are useful.

3. The proposed dataset are valuable for both training and evaluation, and the data curation pipeline using structured, tri-modal annotations (coarse → fine → AV relations, including on/off-screen) seems robust.

**Weaknesses:**

1. Lack of important citations / comparison against prior works V2A / VT2A, including but not limited to [1] - [6]. In particular, the idea of using MLLM to provide conditional signal (text and video) for audio generation is early explored in [1], and the code is fully open-sourced.

2. The architectural design is lack of novelty, basically following very similar structure to MMAudio. Also, ThinkSound already provides very strong CoT reasoning design in its VT2A, what is the key insight that the proposed methodolgy shows unique advantage over ThinkSound? This also seems not clear. So in general, the novelty of the paper remains an important concern.

3. Some compared methods are video-only (e.g., Diff-Foley, Frieren), while ReasonAudio uses video+text. This can inflate CLAP/IB and DeSync. Provide matched “video-only” ablations of ReasonAudio and a “text-only” mode for apples-to-apples.

4. Where is the details of human evaluation? No report of rater count, items per rater, randomization, expert-level of raters. It's important to describe these details to justify the robustness of human eval.

5. Synchrony reward: Precisely how is the synchronization feedback computed? If from Synchformer DeSync, how do you prevent reward hacking and distributional overfitting? Any human-labeled preference data? If it is human-labeled preference data, how sensitive or accurate can human detect the synchronization, is the data quality guaranteed, any auditing procedure applied to ensure it?

[1] Tell What You Hear From What You See - Video to Audio Generation Through Text, NeurIPS 2024.

[2] From Vision to Audio and Beyond: A Unified Model for Audio-Visual Representation and Generation, ICML 2024.

[3] Foleygen: Visually-guided audio generation, 2024.

[4] Video-Guided Foley Sound Generation with Multimodal Controls, CVPR 2024.

[5] SonicVisionLM: Playing Sound with Vision Language Models, CVPR 2024.

[6] Kling-Foley: Multimodal Diffusion Transformer for High-Quality Video-to-Audio Generation.

**Questions:**

See weaknesses.

---

> ### Author Response · Authors · 2025-11-27
>
> ## **Response to Reviewer D3zi (1/2)**
>
> We thank the reviewer for the constructive feedback and for considering our work as "two real pinpoints, proposed method is clear", "$\beta$ ablations are helpful", and "proposed datasets are valuable". We understand that your concerns are mainly related to the paper's model comparison, analysis, and more ablation studies, and we hope our response resolves your concerns fully.
>
>
> **[About the comparison with prior works]**
>
> Thanks for the reviewer’s feedback requesting the comparison with more prior works. We add VATT-Gemma-T [1] and Foleygen [2] for video/text-to-audio generation. The methods are evaluated on the VGGSound test set using perceptual quality, video–audio semantic alignment, and audio–visual temporal synchrony, and we merge them with Table 1 from the main paper as follows:
>
> | Model            | FD (↓) | KL (↓) | IS (↑) | CLAP (↑) | IB (↑) | DeSync (↓) |
> |------------------|-------:|-------:|-------:|---------:|-------:|-----------:|
> | Diff-foley       | 8.29   | 3.15   | 10.8   | 0.12     | 0.19   | 0.81       |
> | Frieren          | **1.34** | 2.53 | 12.3   | 0.19     | 0.22   | 0.89       |
> | V2A-Mapper       | 1.95   | 2.42   | 13.1   | 0.13     | 0.24   | 1.04       |
> | MMAudio          | 1.76   | **1.66** | 13.2 | 0.22     | **0.31** | 0.44     |
> | ThinkSound       | 2.43   | 2.46   | 12.5   | 0.19     | 0.26   | 0.63       |
> | VATT-Gemma-T [1] | 1.64   | 1.95   | 12.8   | 0.22     | 0.26   | 0.77       |
> | Foleygen [2]     | 2.83   | 2.13   | 11.7   | 0.17     | 0.23   | 0.86       |
> | ReasonAudio-Small| 1.89   | 1.80   | **16.9** | _0.23_  | _0.30_ | _0.29_     |
> | ReasonAudio-Large| _1.56_ | _1.75_ | _15.4_ | **0.24** | **0.31** | **0.28** |
>
>
> For temporal alignment, ReasonAudio achieves state-of-the-art synchrony (DeSync $=0.29$), benefiting from preference optimization that explicitly aligns generation with visual synchrony preferences. For semantic alignment, ReasonAudio shows stronger cross-modal coherence, with an ImageBind score of $0.30$ (video–audio) and a CLAP score of $0.23$ (text–audio). In contrast, baselines with MLLM (Thinksound and VATT-Gemma-T [1]) necessitate tuning LLMs on video understanding and subsequently training the audio generator, naturally posing challenges from LLM overfitting and multi-stage training. In contrast, ReasonAudio uses learnable queries to directly bridge multimodal understanding and audio generation within a unified framework, strengthening end-to-end multimodal reasoning.
>
> [1] Tell What You Hear From What You See - Video to Audio Generation Through Text.
>
> [2] Foleygen: Visually-guided audio generation.
>
> **[About the comparison over ThinkSound and MMAudio]**
>
> Thanks for the reviewer's feedback that requests more explanation on the technical comparison with previous works:
>
> - **Compared with ThinkSound.** Recent work has explored extending the success of MLLMs to multimodal diffusion generation. ThinkSound fine-tunes MLLMs to produce reasoning chains that explicitly model temporal dependencies and decompose audio editing events. It necessitates tuning LLMs on video understanding and subsequently training the audio generator, naturally posing challenges from LLM overfitting and multi-stage training. \textbf{In contrast, ReasonAudio avoids multi-stage MLLM fine-tuning by using learnable queries to directly bridge multimodal understanding and audio generation within a unified framework}, strengthening end-to-end multimodal reasoning. With this design, ReasonAudio achieves strong cross-modal coherence, reaching an ImageBind score of $0.30$ (video--audio) and a CLAP score of $0.23$ (text--audio).
> - **Compared with MMAudio.** We agree with the reviewers that we adopt an MM-DiT backbone to jointly model video, audio, and text within a unified transformer, which is consistent with the practice in MMAudio and ThinkSound. However, the major challenge for VT2A generation is achieving robust temporal alignment, which remains a persistent bottleneck in MMAudio. In ReasonAudio, we perform preference post-training (DPO, RWR, SFT) using synchrony feedback, aligning the generator with prior knowledge and preferences on visual–audio synchrony. Experimental results demonstrate that ReasonAudio attains state-of-the-art synchrony with DeSync $=0.29$.

---

> > ### Author Response · Authors · 2025-11-27
> >
> > ## **Response to Reviewer D3zi (2/2)**
> >
> > **[About the video-only and text-only comparison]**
> >
> > Thanks for the reviewer's suggestions on the video-only and text-only comparison with prior works. ReasonAudio is capable of video-to-audio synthesis (video-only mode) or text-to-audio synthesis (video-only mode) without additional fine-tuning.
> >
> > - **Video-only.** For video-to-audio (V2A) generation, we use the VGGSound test set as the standard benchmark and evaluate ReasonAudio-Small in terms of perceptual quality, video–audio semantic alignment, and audio–visual temporal synchrony. As shown in Table, ReasonAudio under pure V2A conditioning achieves strong cross-modal coherence, with an ImageBind score of $0.24$ and a DeSync score of $0.29$, while maintaining comparable perceptual quality (FD) to competitive baselines.
> > - **Text-only.** We presented the text-only comparison in Table 2 in the main paper, where we adopt the AudioCaps test set as the standard benchmark and compare ReasonAudio with TANGO 2, Make-An-Audio 2, SoundCTM, and MMAudio.
> >
> >
> > | Model            | FD (↓) | KL (↓) | IS (↑) | IB (↑) | DeSync (↓) |
> > |------------------|-------:|-------:|-------:|-------:|-----------:|
> > | ReasonAudio-VT2A | 1.89   | 1.80   | 16.9   | 0.23   | 0.29       |
> > | ReasonAudio-V2A  | 1.91   | 1.83   | 16.7   | 0.24   | 0.29       |
> >
> > **[About the details of human evaluation]**
> >
> > Thanks to the reviewer for raising suggestions about the details of human evaluation. We provide the full setup in Appendix D. Ratings are collected independently for model-generated samples and reference audio, and we recruit 20 raters with normal hearing. All samples (50 video-audio pairs per subject score) are presented in randomized order to mitigate ordering effects. We report each subjective metric as mean $\pm$ standard deviation (SD) in the main paper to reduce randomness, where SD reflects the variability of ratings across samples and raters.
> >
> >
> >
> > **[About the details of synchronization feedback]**
> >
> > Thanks for the reviewer's feedback that requests more explanation on the usage of synchronization feedback. We leverage an audio–visual temporal alignment classifier [3] as the reward model: given an audio–video pair $(x_0, y)$, the classifier predicts an alignment score ($\uparrow$), and we define the reward as $r(x_0,y)=\textbf{Align}(x_0,y)$. For evaluation, we report the synchronization score (DeSync) ($\downarrow$) from Synchformer, which measures audio–video misalignment and is distinct from our reward signal. Using a different model/metric for evaluation reduces the risk of reward hacking or overfitting to the reward model’s distribution.
> >
> > To train the classifier, [3] uses three types of pairs: 50\% of the pairs are real audio-visual pairs (true pair) labeled as 1, 25\% are audio-visual pairs from the same video but temporally shifted (temporal shift pair) labeled as 0, and the last 25\% are audio-visual pairs from different videos (wrong pair) labeled as 0. The resulting alignment classifier achieves 90\% test accuracy, indicating that it provides a meaningful signal for audio–visual synchronization, and it has also been used for evaluation in related work such as VATT-Gemma-T [1].
> >
> > [3] Diff-Foley: Synchronized Video-to-Audio Synthesis with Latent Diffusion Models.
> >
> >
> > **Again, we appreciate the reviewer's valuable reviews and believe your concerns are mainly due to the paper's model comparison, analysis, and more ablation studies. In this rebuttal phase, we will try to address them carefully, and we would be grateful if you could update your score if our responses resolve your concerns.**

---

### Official Review · Reviewer_2MVJ · 2025-11-01

**Soundness:** 2
**Presentation:** 2
**Contribution:** 2
**Rating:** 4
**Confidence:** 4

**Summary:**

The paper presents ReasonAudio, a multimodal reasoning framework for video–text-to-audio (VT2A) generation. It integrates a frozen MLLM (Qwen-VL-2.5) with learnable semantic queries, a triple-stream MMDiT architecture (semantic, temporal, audio), and a flow-matching generation backbone. A new dataset, VGGSound-Think, is introduced, featuring structured annotations with `<HINT>` and `<TRACK>` elements to enhance semantic and temporal alignment.
 Preference-based training (Flow-DPO / Flow-RWR) is employed to improve synchrony (DeSync) without heavily sacrificing audio fidelity. The system achieves strong results on VGGSound and AudioCaps benchmarks, showing promising capabilities in semantic reasoning and synchronization.

**Strengths:**

1. **Clear and polished writing.**
    The paper is well-structured and easy to follow. The motivation, methodology, and experimental setup are clearly connected, and the figures (especially Fig.1 and Fig.3) effectively communicate the conceptual framework.
2. **Use of MLLM brings insightful capabilities.**
    The paper demonstrates that MLLMs contribute to long-range, cross-modal understanding. The model can infer implicit sound sources (e.g., sirens from “calling 911”) and better handle multi-object, on/off-screen semantics compared to CLIP-only or text-only baselines.
3. **Quantitative and qualitative improvements.**
    The results on VGGSound and AudioCaps show clear gains in semantic alignment (CLAP/IB) and synchrony (DeSync), confirming the system’s overall effectiveness.

**Weaknesses:**

**1. Dataset construction and contribution clarity**

The dataset creation process is insufficiently described. The authors state that captions and relationships are generated using **GPT-4o**, but no details are given about how GPT-4o was prompted, filtered, or validated.
 From practical experience, GPT-4o’s **audio and video understanding is limited**, particularly for complex, real-world VGGSound clips involving multiple overlapping sources or off-screen events. Therefore, it is unclear whether this dataset truly advances beyond existing captioning sets like *Sound-VECaps*.

To justify the claimed data contribution, the authors should:

- Quantitatively verify dataset quality (e.g., inter-annotator consistency, caption diversity, and alignment accuracy);
- Compare with existing datasets on sound-event richness and relational structure;
- Provide ablation experiments isolating the effect of dataset vs. model design.
   At present, improvements may come from model training rather than data innovation.

**2. Choice of MLLM backbone**

The choice of **Qwen-VL-2.5** instead of **Qwen-2.5-Omni** is questionable. Qwen-VL’s instruction tuning emphasizes **image-related comprehension**, potentially weakening its textual-audio reasoning ability.
 An additional experiment using Qwen-2.5-Omni (with the same size and configuration) would clarify whether the gains arise from multimodal reasoning or just visual prior knowledge. Also, the paper inconsistently refers to both “Qwen-VL-7B-Instruct” and “Qwen-VL-8B”—this should be unified and clarified.

**3. DPO as a contribution**

The use of **DPO** for preference alignment appears straightforward. The paper itself admits that DPO improves synchronization (DeSync) but slightly degrades FD. Without deeper analysis or novel adaptation, this seems more like an application of existing methodology.
 A more detailed **trade-off analysis**  or a multi-objective optimization variant would strengthen the originality claim.

**4. Experimental transparency and missing baselines**

- Training setup lacks detail, achieving the reported performance from 200k iterations without pretraining seems unlikely.
- The MLLMs-for-Understanding section omits a CLIP+LLM hybrid baseline, which would clarify the specific contribution of MLLMs beyond visual-text fusion.
- Comparisons on VGGSound-Think against *DeepSound-V1* and *ThinkSound* are not meaningful, since those baselines were never trained on this data format. Results may simply reflect dataset mismatch.
- Input formats (<HINT> vs <TRACK>) are not explicitly described per task (VT2A, T2A, reasoning), making replication difficult.

**5. Reasoning validation and robustness**

- Reasoning ability is mostly supported by qualitative samples rather than quantitative verification.

**6. Robustness**

- Additionally, robustness under weak or missing textual inputs (e.g., pure V2A mode) should be tested, as the system’s dependency on structured prompts may limit real-world usability.

**7. Novelty**

- Treating understanding as a preceding stage of generation is intuitively reasonable and valid; however, this stage primarily operates at the prompt level, which does not fundamentally enhance the diffusion model’s intrinsic understanding ability, thereby limiting its originality.

**Questions:**

See Weaknesses

If the authors provide clearer dataset validation, justify MLLM choice, and strengthen the DPO and reasoning analyses, I would be happy to discuss.

---

> ### Author Response · Authors · 2025-11-27
>
> ## **Response to Reviewer 2MVJ (1/4)**
>
> We thank the reviewer for the constructive feedback and for considering our work as "use of LLM brings insightful capabilities", "quantitative and qualitative improvements", and "clear and polished writing". We understand that your concerns are mainly related to the paper's dataset validation, model analysis, and more ablation studies, and we hope our response resolves your concerns fully.
>
>
> **[About the details of data construction with GPT]**
>
> Thanks for the reviewer's feedback that requests more explanation on the details of data construction with GPT. Each sample is annotated through a structured, step-by-step procedure:
>
>
> - **Coarse-grained acoustic gist.** For each audio clip, we prompt GPT-4o with:
>
>   > You are given a video (frames) and its audio. Write a *single-sentence* coarse acoustic gist that summarizes the dominant sound events and overall ambience *without naming specific sound-producing objects* (e.g., avoid "car", "dog", "siren"). Instead, describe sound *attributes* such as pitch, timbre, rhythm, intensity, continuity, and background/foreground. Output only the one sentence.
>
>   to obtain a high-level hint that captures the main acoustic content while remaining object-agnostic.
>
> - **Fine-grained sound grounding.** We prompt GPT-4o with:
>
>   > You are given a video (frames) and its audio. Write **one concise paragraph** (2–4 sentences) that **grounds salient sounds to visible entities** and describes how the sound evolves over time.
>   >
>   > **Requirements:**
>   > (1) Mention only entities that are clearly supported by the video (do NOT invent objects).
>   > (2) Explicitly connect each salient sound to its most likely on-screen source; if the source is likely off-screen, say so.
>   > (3) Include temporal progression using natural phrasing (e.g., "begins", "rises", "fades", "as X happens").
>   > (4) Prefer concrete audio descriptors (e.g., "wailing siren", "engine chugging", "faint speech", "reverberant") and visual evidence cues (e.g., "a truck is shown", "mouth movement", "vehicle starts moving").
>
>   to align sound events with specific visible objects, refining the gist into grounded descriptions.
>
> - **Structured audio–visual relation annotations.** For each grounded object, we prompt GPT-4o with:
>
>   > You are given a video (frames) and its audio. Produce a **structured, line-based annotation** of the main sound-producing entities.
>   >
>   > **Step 1: Identify objects.**
>   > List 2–6 entities that plausibly produce salient sounds. Name each entity with a short **snake_case** identifier (e.g., `mechanical_siren`, `human_speech`, `fire_truck_engine`). Use only entities supported by the video.
>   >
>   > **Step 2: For each object, output exactly ONE line in the following format:**
>   > `<Object '{object_id}' screen={on_screen/off_screen}> {audio_keywords}`
>   >
>   > **audio_keywords requirements:**
>   > - Provide 3–8 terse keywords/phrases separated by spaces.
>   > - Focus on sound attributes and events, e.g., "wailing", "rise_fall", "reverberant", "faint", "rumble", "chugging", "idling".
>   > - If helpful, include a single interaction cue such as "masked_by_X" or "overlaps_with_X" (where X is another "object_id").
>
>   to generate descriptive keywords and explicit audio–visual relations, including on-/off-screen attribution and interactions among objects.
>
> - **Filtering and validation.** We conduct a strict manual check where 5\% of the full dataset is randomly sampled for human review as a quality control measure. This procedure serves as a check on our filtering rules and helps ensure the dataset’s overall reliability. During inspection, reviewers evaluate both technical alignment (e.g., audio–visual synchronization and consistency) and perceptual correspondence between the sound and the visual content.
>
> **[About ablation isolating the effect of dataset vs. model design.]**
>
> Thanks for the reviewer’s feedback requesting more ablation isolating the effect of the dataset vs. the model design. We leverage the VGGSound and VGGSound-Think as the training set to guarantee the same model is used and isolate the effect of the dataset on generation quality. We adopt the VGGSound test set as a standard benchmark and evaluate ReasonAudio-Small in terms of perceptual quality (FD, KL) and text–audio semantic alignment (CLAP). With the same model architecture, ReasonAudio trained on VGGSound-Think achieves enhanced perceptual quality (FD $2.36$, IS $11.7$) and stronger text–audio semantic alignment with the CLAP score of $0.34$, showcasing the benefits of learning semantically rich textual descriptions in the VGGSound-Think dataset.
>
> | Trainset        | FD (↓) | KL (↓) | IS (↑) | CLAP (↑) |
> |----------------|-------:|-------:|-------:|---------:|
> | VGGSound       | 2.41   | 1.45   | 11.4   | 0.33     |
> | VGGSound-Think | **2.36** | **1.43** | **11.7** | **0.34** |

---

> > ### Author Response · Authors · 2025-11-27
> >
> > ## **Response to Reviewer 2MVJ (2/4)**
> >
> > **[About dataset quality and sound-event richness.]**
> >
> > Thanks for the reviewer’s feedback requesting more explanation and comparison of the data quality. We use metrics (mean pairwise cosine distance, VLM-as-Judge) to evaluate caption diversity and alignment accuracy.
> >
> > - **Caption diversity.** We randomly select 10 video classes (e.g., *baby*, *fireworks*), then sample 20 captions per class and compute the *mean pairwise cosine distance* between their T5 embeddings (higher indicates more diverse phrasing/semantics within the class).
> >
> > - **Alignment accuracy.** We randomly sample 5% of the full dataset and ask a VLM-as-judge to perform pairwise preference comparisons: given the same video/audio pair and two candidate captions (from VGGSound vs. VGGSound-Think), the judge selects the caption with better audio–visual alignment (“win”). To reduce potential bias from the caption construction process, we use Gemini-3 as an external judge model, rather than the model used for data generation.
> >
> > | Data            | Caption diversity (↑) | Alignment accuracy (↑) |
> > |----------------|----------------------:|------------------------:|
> > | VGGSound       | 0.51                  | 37.4%                   |
> > | VGGSound-Think | 0.87                  | 62.6%                   |
> >
> > As can be seen in the Table, VGGSound-Think exhibits distinctly higher caption diversity and alignment accuracy than VGGSound. In VGGSound, the text annotations are often sparse and semantically shallow. In contrast, VGGSound-Think enriches VGGSound with semantically informative descriptions, providing a stronger foundation for reasoning over acoustic cues and structured audio–visual relationships.
> >
> >
> > **[About the MLLM backbone]**
> >
> > Thanks for the reviewer’s feedback requesting more details about the MLLM backbone. We built our understanding module on the open-source Qwen2.5-VL-7B-Instruct, as it natively supports text and video inputs, and aligns closely with our setting on text-video-to-audio generation.
> >
> > Following the reviewer’s suggestion, we ablate the backbone by replacing it with Qwen2.5-Omni-7B-Instruct and test in the VGGSound-Think test set. As can be seen in Table, the Omni-based ReasonAudio presents improvement in fidelity (FD) and slight advancement in text-audio alignment (CLAP), while witnessing a degradation in video alignment (IB). To conclude, Omni model is trained with video–audio supervision and is therefore stronger at tri-modal (text–video–audio) reasoning and alignment. At the same time, its visual understanding capacity appears slightly worse than the VL variant in our setting, which likely contributes to the reduced video alignment. We will add this discussion to the revised version of the paper.
> >
> >
> > | Model     | FD (↓) | CLAP (↑) | IB (↑) |
> > |----------|-------:|---------:|-------:|
> > | Qwen-VL  | 2.09   | 0.28     | 0.32   |
> > | Qwen-Omni| 1.98   | 0.29     | 0.31   |
> >
> > Thanks for the reviewer's reminder about the typo on Qwen2.5-VL-7B-Instruct, and we will refine the mentioned instance in the revised version of the paper.

---

> > > ### Author Response · Authors · 2025-11-27
> > >
> > > ## **Response to Reviewer 2MVJ (3/4)**
> > >
> > > **[About the more detailed DPO trade-off analysis]**
> > >
> > > Thanks for the reviewer's feedback that requests more explanation on audio quality and synchronization trade-off in DPO post-training. In DPO post-training, $\beta$ is leveraged as a hyperparameter to control the trade-off between the strength of the policy update and distance to the pretrained model. Increasing $\beta$ accelerates reward improvement, but beyond $\beta=5$ we observe a noticeable drop in audio quality (higher FD), indicating overfitting toward the reward model. Although the alignment–fidelity trade-off is observed, 1) ReasonAudio-Large ranks the second-best performance model in audio quality (FD), and 2) ReasonAudio-Small and ReasonAudio-Large attain state-of-the-art performance with an IS score of $11.7, 11.5$
> > >
> > > To make an in-depth evaluation, the alignment–fidelity trade-off we observe persists prior reward-finetuning work:
> > > - 1) In audio generation, Tango 2 [1, Table 3] reports that the DPO model attains higher CLAP similarity (from 0.54 to 0.57) compared to the pretrained model, but with a higher FAD (from 2.51 to 2.69), indicating an FAD–CLAP trade-off under reward fine-tuning;
> > > - 2) In music generation [2, Table 2], the GRPO post-trained model improves content enjoyment (CE) while slightly reducing production quality (PQ), again illustrating a trade-off between content alignment and perceptual fidelity. We will analyze and explain more about this trade-off in the revised version of the paper.
> > >
> > > To balance adaptively during post-training, one alternative way is to dynamically calibrate $\beta$ at data quality considerations [3], where $\beta$ is adaptively decreased for closely-matched pairwise data (i.e., low gap data) to facilitate assertive updates, and increased for easily-discriminated pairs (i.e., high-gap data), preventing overfitting to noise. It demonstrates that the dynamic $\beta$ adjustment technique offers a more robust and adaptable training paradigm for aligning models with human feedback.
> > >
> > > [1] Tango 2: Aligning Diffusion-based Text-to-Audio Generations through Direct Preference Optimization.
> > >
> > > [2] Towards Hallucination-Free Music: A Reinforcement Learning Preference Optimization Framework for Reliable Song Generation.
> > >
> > > [3] $\beta$-DPO: Direct Preference Optimization with Dynamic $\beta$
> > >
> > >
> > > **[About the details of training steps]**
> > >
> > > Thanks for the reviewer’s request for additional training details. In our main experiments, we train with a batch size of 512 for 200K optimization steps, followed by 100K steps of preference-based post-training. With this large batch size, the full training run converges in approximately 36 hours. We will include these details in the revised version of the paper.
> > >
> > > **[About the CLIP + LLM Hybrid baseline]**
> > >
> > > Thanks for the reviewer’s suggestion on the CLIP + LLM Hybrid baseline comparison. Following the suggestion, we construct a hybrid pipeline that uses CLIP for visual understanding and an LLM for reasoning text. As can be seen in Table, this hybrid baseline achieves comparable text–audio alignment (CLAP), indicating that an LLM can effectively reason over dynamic, long-range context comparable to MLLMs. However, the CLIP+LLM hybrid yields lower audio–visual semantic alignment (IB), suggesting that replacing an MLLM with a CLIP-based visual encoder weakens cross-modal understanding. It highlights the advantage of MLLMs as the multimodal understanding component, which better integrates visual cues with textual reasoning, strengthening audio–visual reasoning, and generates audio semantically aligned with the input video.
> > >
> > > | Model    | FD (↓) | CLAP (↑) | IB (↑) |
> > > |---------|-------:|---------:|-------:|
> > > | MLLM    | 2.09   | 0.28     | 0.32   |
> > > | CLIP+LLM| 2.06   | 0.28     | 0.28   |
> > > | CLIP    | 1.92   | 0.24     | 0.25   |

---

> > > > ### Author Response · Authors · 2025-11-27
> > > >
> > > > ## **Response to Reviewer 2MVJ (4/4)**
> > > >
> > > > **[About the Reasoning and DeepSound and ThinkSound baselines]**
> > > >
> > > > Thanks for the reviewer’s feedback requesting an explanation on the DeepSound and ThinkSound comparison. Our reasoning captions are expressed in natural language and include three components: coarse-grained acoustic gist, fine-grained sound grounding, and structured audio–visual relation annotations. As baselines, DeepSound and ThinkSound directly rely on textual representations by fine-tuning MLLMs to predict captions before connecting to diffusion models, and thus data format mismatch might exist on the LLM side but not happen on the diffusion side.
> > > >
> > > > | Model      | FD (↓) | CLAP (↑) | IB (↑) |
> > > > |-----------|-------:|---------:|-------:|
> > > > | ThinkSound| 2.49   | 0.24     | 0.30   |
> > > > | ReasonAudio | 2.09 | 0.28     | 0.32   |
> > > >
> > > > To further mitigate concerns about such a mismatch, we train ThinkSound using the VGGSound-Think annotations, ensuring that the text conditioning is consistent with the ReasonAudio training procedure. As shown in Table, ReasonAudio achieves stronger text–audio semantic alignment (CLAP: $0.28$) and more robust video–audio coherence (ImageBind: $0.32$), suggesting the effectiveness of learnable prompts derived from strong MLLMs, which provide robust multimodal understanding and serve as an effective bridge between understanding and generation.
> > > >
> > > > **[About the robustness and video-only inference]**
> > > >
> > > > Thanks for the reviewer’s feedback on the pure V2A inference. For video-to-audio (V2A) generation, we use the VGGSound test set as the standard benchmark and evaluate ReasonAudio-Small in terms of perceptual quality, video–audio semantic alignment, and audio–visual temporal synchrony. As shown in Table, ReasonAudio under pure V2A conditioning achieves strong cross-modal coherence, with an ImageBind score of $0.24$ and a DeSync score of $0.29$, and maintains comparable perceptual quality (FD) to competitive baselines.
> > > >
> > > > In summary, while our structural prompts explicitly provide acoustic hints and structured audio–visual relation descriptions (e.g., object grounding and on-/off-screen attribution), the model remains robust even without text guidance for two reasons:
> > > > - 1) Instead of directly relying on textual representations, ReasonAudio conditions on learnable prompts derived from strong MLLMs, which provides robust multimodal understanding and serves as an effective bridge between understanding and generation;
> > > > - 2) Besides, we apply random conditional feature dropping during classifier-free guidance (CFG) training, which improves cross-modal generalization while preserving the flexibility of using an text input.
> > > >
> > > > | Model           | FD (↓) | KL (↓) | IS (↑) | IB (↑) | DeSync (↓) |
> > > > |----------------|-------:|-------:|-------:|-------:|-----------:|
> > > > | ReasonAudio-VT2A | 1.89  | 1.80   | 16.9   | 0.23   | 0.29       |
> > > > | ReasonAudio-V2A  | 1.91  | 1.83   | 16.7   | 0.24   | 0.29       |
> > > >
> > > >
> > > > **Again, we appreciate the reviewer's valuable reviews and believe your concerns are mainly due to the paper's dataset validation, model analysis, and more ablation studies. In this rebuttal phase, we will try to address them carefully, and we would be grateful if you could update your score if our responses resolve your concerns.**

---

### Official Review · Reviewer_6u3X · 2025-11-02

**Soundness:** 2
**Presentation:** 2
**Contribution:** 2
**Rating:** 4
**Confidence:** 4

**Summary:**

The core goal of ReasonAudio is to synthesize realistic ambient sounds conditioned on both video and text, achieving superior semantic alignment and temporal synchronization.

Key Contributions: New Reasoning Dataset (VGGSound-Think): A tri-modal (video-text-audio) dataset augmenting $\text{VGGSound}$ with semantically rich annotations, including acoustic hints and structured audio-visual relation descriptions (e.g., multi-object interactions, on-/off-screen attribution).MLLM-Empowered Semantic Understanding: A novel approach that leverages frozen Multimodal Large Language Models ($\text{MLLMs}$) and learnable queries to bridge the $\text{MLLM}$'s understanding component with the flow-matching generative component, avoiding complex multi-stage training.Temporal Alignment via Preference Optimization: Application of preference post-training techniques ($\text{Flow-DPO}$, $\text{Flow-RWR}$) using synchronization feedback to align the generative model with visual synchrony preferences.

**Strengths:**

The two key solutions are highly original. The use of learnable queries to integrate a frozen $\text{MLLM}$ (like $\text{Qwen2.5-VL-8B}$) directly into the flow-matching pipeline for deep semantic reasoning is a novel simplification over prior multi-stage approaches[cite: 155, 321, 681]. Furthermore, applying preference optimization ($\text{DPO}/\text{RWR}$) with synchronization feedback to explicitly boost temporal alignment is a unique and effective strategy for $\text{VT2A}$

The construction of VGGSound-Think is a high-quality contribution, focusing on semantically rich, multi-grained annotations essential for reasoning. The ablation study confirms the efficacy of the proposed components: the $\text{MLLM}$ (vs. $\text{CLIP}$ or $\text{LLM}$) significantly improves semantic scores, and $\text{DPO}$ consistently yields the best temporal alignment ($\text{DeSync}$). The architecture, based on flow-matching and $\text{MM-DiT}$, is state-of-the-art.

The paper clearly articulates the two major challenges ($\text{semantic}$ and $\text{temporal}$ gaps) and presents a targeted solution for each. The architecture (Figure 3) and the $\text{VGGSound-Think}$ annotation steps (Figure 2) are well-illustrated and easy to follow. The mathematical formulations for $\text{SFT}$, $\text{RWR}$, and $\text{DPO}$ loss are clearly presented in the context of flow-matching.

ReasonAudio achieves state-of-the-art $\text{VT2A}$ performance, with notable gains in semantic alignment ($\text{CLAP}$ and $\text{IB}$) and a substantial improvement in temporal synchronization ($\text{DeSync}$). The case studies demonstrate a critical capability: reasoning over acoustic hints and on-/off-screen attribution. This work sets a new benchmark for multimodal audio synthesis by moving beyond shallow text captions to deep, nuanced reasoning.

**Weaknesses:**

Preference Optimization vs. Audio Fidelity Trade-off: The ablation study shows that preference post-training introduces a slight degradation in audio quality (higher $\text{FD}$)4. Specifically, DPO with $\beta=5$ provides the best synchrony but increasing $\beta$ beyond this value causes a "noticeable drop in audio quality (higher $\text{FD}$)," indicating overfitting to the synchrony reward. Suggestion: A deeper investigation into a principled way to balance the temporal synchrony reward against the audio fidelity loss ($\mathcal{L}_{\text{FM}}$) is needed, perhaps by making the fidelity term adaptive during post-traini

Limited Exploration of Learnable Query Design: The paper states $\text{N}=77$ learnable query tokens were used. While the concept is novel, the selection and impact of this hyperparameter ($N$) are not ablated. The choice of $N=77$ (the same length as the CLIP token limit mentioned earlier 6) suggests a potential link that should be justified or explored. Suggestion: The authors should include an ablation study on the number of learnable queries ($N$) to demonstrate its optimal configuration and effect on the MLLM's reasoning capacity and model complexity.

Dependence on GPT-4o: The creation of the rich VGGSound-Think dataset heavily relies on $\text{GPT-4o}$ for Foley Reasoning Caption Generation7. While $\text{GPT-4o}$ is state-of-the-art, this reliance means the quality and type of reasoning generated (e.g., acoustic hints, off-screen attribution) are fundamentally limited by the capabilities and potential biases of a closed-source model. Suggestion: A brief discussion on the generalizability of $\text{VGGSound-Think}$ annotations, specifically how performance might change if a different MLLM (e.g., a fully open model) were used for the annotation pipeline, would strengthen the contribution.

**Questions:**

Synchronization Reward Function: The paper mentions using synchronization feedback from $\text{Synchformer}$ to define the preferences/rewards for $\text{DPO}$ and $\text{RWR}$. Could the authors explicitly define the reward function $r(x_0, y)$? Is the reward derived directly from the negative $\text{DeSync}$ score, or is it a more complex function incorporating $\text{DeSync}$ score and other metrics (e.g., $\text{IB}$ score) to reflect overall video-audio fit?

MLLM Inference Latency/Overhead: While the $\text{MLLM}$ is frozen, generating the semantic conditions still adds computation time compared to simple fixed-feature encoders ($\text{CLIP}$). Could the authors provide a comparison of the inference latency or computational overhead introduced by the frozen $\text{MLLM} + \text{learnable queries}$ pipeline versus the purely $\text{CLIP}$-based baseline? This is crucial for evaluating the practical efficiency of ReasonAudio.

Generalization to Novel Concepts: The $\text{VGGSound-Think}$ set focuses on refining existing $\text{VGGSound}$ concepts. Does the $\text{MLLM}$-empowered approach demonstrate superior zero-shot generalization to completely novel video/audio concepts that were not represented in the original $\text{VGGSound}$ or the original $\text{Qwen2.5}$ training data? A qualitative example or a metric on the $\text{Movie Gen Audio Bench}$ dataset that highlights this capability would be highly valuable.

---

> ### Author Response · Authors · 2025-11-27
>
> ## **Response to Reviewer 6u3X (1/2)**
> We thank the reviewer for the constructive feedback and for considering our work as "key contribution highly original", "VGGSound-Think is a high quality contribution", and "clearly addresses two major challenges". We understand that your concerns are mainly related to the paper's analysis and more ablation studies, and we hope our response resolves your concerns fully.
>
> **[About balancing audio quality and reward alignment]**
>
> Thanks for the reviewer's feedback that requests more explanation on audio quality and reward alignment. In DPO post-training, $\beta$ is leveraged as a hyperparameter to control the trade-off between the strength of the policy update and distance to the pretrained model, where beyond $\beta=5$ we observe a noticeable drop in audio quality (higher FD) due to overfitting towards the reward model.
>
> To balance adaptively during post-training, one alternative way is to dynamically calibrate $\beta$ at data quality considerations [1], where $\beta$ is adaptively decreased for closely-matched pairwise data (i.e., low gap data) to facilitate assertive updates, and increased for easily-discriminated pairs (i.e., high-gap data), preventing overfitting to noise. It demonstrates that the dynamic $\beta$ adjustment technique offers a more robust and adaptable training paradigm for aligning models with human feedback.
>
> [1] $\beta$-DPO: Direct Preference Optimization with Dynamic $\beta$
>
>
> **[About the number of learnable queries]**
>
> Thanks for the reviewer's feedback that requests more clarification on the number of learnable queries. For the MLLM understanding module, we use $N=77$ learnable queries intended to enable a fair comparison between LLM and CLIP-based conditioning by matching the representation shape $\mathcal{Q} \in \mathbb{R}^{N \times D}$, where we use $D$ equals the MLLM hidden dimension.
>
> Following the reviewer’s suggestion, we ablate the understanding module on the VGGSound-Think test set by varying the number of learnable queries. As can be seen in Table, reducing the number of queries consistently degrades semantic scores (CLAP and IB). Increasing $N$ accelerates improvement, but gains saturate at $N=128$, where we observe only marginal improvements, indicating that learnable queries effectively compress conditioning information into a fixed-length token set, which aligns with the findings from Emu2 [2] that each image regardless of its resolution can be encoded into a fixed length of 64 continuous vectors, providing both compact and semantically rich latent embeddings.
>
>
> | Queries | FD (↓) | CLAP (↑) | IB (↑) |
> |--------:|-------:|---------:|-------:|
> | 128     | 2.04   | 0.29     | 0.33   |
> | 77      | 2.09   | 0.28     | 0.32   |
> | 64      | 2.17   | 0.26     | 0.31   |
> | 32      | 2.32   | 0.24     | 0.29   |
>
> [2] Generative multimodal models are in-context learners.
>
> **[About the dependence on GPT-4o]**
>
> Thanks to the reviewer for raising concerns about our dependence on GPT-4o. It provides foundations for audio understanding, captioning, and segmentation tasks. Following the suggestion, we evaluate and compare with VideoLLaMA3 [3], an open-source MLLM, to extract temporal and semantic information from videos. Using the same example in Figure 2 ("fire truck siren"), we provide a qualitative comparison in the table.
>
> |  | **GPT-4o** | **VideoLLaMA3** |
> |---|---|---|
> | **Fine-grained sound grounding** | The scene is an urban street or parking lot at dusk, featuring a vintage red fire truck.<br>A mechanical siren begins, winding up and then rapidly down in pitch.<br>As the siren fades, a faint voice and the low, chugging rumble of the truck's engine becomes prominent as the vehicle slowly begins to drive away. | A fire truck drives on a city street.<br>The siren is loud and varies in pitch, with engine/road noise underneath.<br>After it passes, the siren weakens and background/engine noise remains. |
> | **Structured audio--visual relations** | `<Object mechanical_siren' screen=on_screen>` wailing rise_fall; reverberant<br>`<Object human_speech' screen=on_screen>` faint; brief; reverberant<br>`<Object antique_fire_truck_engine' screen=on_screen>` idling rumble; chugging | fire truck (red), flashing lights<br>siren: loud, up/down pitch, then quieter<br>engine/road noise present |
>
>
> VideoLLaMA3 demonstrates strong high-level multimodal understanding and conversational ability. However, it is less efficient at following our instruction format: (1) for fine-grained sound grounding, it tends to produce shorter and less precise descriptions; and (2) for structured audio–visual relation annotations, its outputs contain fewer explicit fields and are less consistently structured.
>
> [3] VideoLLaMA 3: Frontier Multimodal Foundation Models for Image and Video Understanding.

---

> > ### Author Response · Authors · 2025-11-27
> >
> > ## **Response to Reviewer 6u3X (2/2)**
> >
> > **[About the reward function]**
> >
> > Thanks for the reviewer's feedback that requests more explanation on the usage of synchronization feedback. We leverage an audio–visual temporal alignment classifier [3] as the reward model: given an audio–video pair $(x_0, y)$, the classifier predicts an alignment score ($\uparrow$), and we define the reward as $r(x_0,y)=\textbf{Align}(x_0,y)$. For evaluation, we report the synchronization score (DeSync) ($\downarrow$) from Synchformer, which measures audio–video misalignment and is distinct from our reward signal. Using a different model/metric for evaluation reduces the risk of reward hacking or overfitting to the reward model’s distribution.
> >
> > To train the classifier, [4] uses three types of pairs: 50\% of the pairs are real audio-visual pairs (true pair) labeled as 1, 25\% are audio-visual pairs from the same video but temporally shifted (temporal shift pair) labeled as 0, and the last 25\% are audio-visual pairs from different videos (wrong pair) labeled as 0. The resulting alignment classifier achieves 90\% test accuracy, indicating that it provides a strong signal for audio–visual synchronization, and it has also been used for synchrony evaluation in related work such as VATT-Gemma-T [5].
> >
> > [4] Diff-Foley: Synchronized Video-to-Audio Synthesis with Latent Diffusion Models.
> >
> > [5] Tell What You Hear From What You See - Video to Audio Generation Through Text.
> >
> > **[About the MLLM inference latency]**
> >
> > Thanks for the reviewer’s suggestion to provide a more detailed analysis of model latency. Following the reviewer's suggestion, we report latency using the real-time factor (RTF) between MLLM and CLIP based models, defined as the ratio between the processing time of an utterance and its duration (lower is better):
> >
> > | Model | RTF (↓) | FD (↓) | CLAP (↑) | IB (↑) |
> > |------|--------:|-------:|---------:|-------:|
> > | MLLM | 0.27    | 2.09   | 0.28     | 0.32   |
> > | CLIP | 0.13    | 1.92   | 0.24     | 0.25   |
> >
> > It can be seen that MLLM introduces only a modest and acceptable computational overhead relative to the CLIP-based approach, making it practical for real-time applications. Importantly, the MLLM with learnable prompts produces conditioning latents in a single forward pass without requiring autoregressive decoding steps, significantly improving generation efficiency and avoiding LLM overfitting or multi-stage training.
> >
> > **[About generalization to novel concepts]**
> >
> > Thanks for the reviewer’s suggestion to provide a more thorough analysis of generalization to novel concepts. To assess generalization, we include additional qualitative visualizations of video-to-audio generation on the MovieGen Audio benchmark in Appendix E and on our demo page. Following the reviewer’s suggestion, we also report objective metrics (IS, IB, CLAP, and DeSync score) to quantify fidelity and alignment.
> >
> > | Method      | IS (↑) | IB (↑) | CLAP (↑) | DeSync (↓) |
> > |------------|-------:|-------:|---------:|-----------:|
> > | MMAudio    | 8.40   | 27.0   | 0.43     | 0.77       |
> > | ReasonAudio| 8.96   | 32.8   | 0.46     | 0.59       |
> >
> >
> > Compared to MMAudio as baseline, ReasonAudio achieves strong text–audio semantic alignment with a CLAP score of $0.46$ and robust video–audio coherence with an ImageBind score of $32.8$, indicating better generalization to the MovieGen Audio benchmark and improved adherence to semantic conditioning. These results highlight the advantage of using MLLMs as the understanding module: Compared to the CLIP encoder, they generalize more effectively and provide stronger semantic reasoning.
> >
> >
> >
> > **Again, we appreciate the reviewer's valuable reviews and believe your concerns are mainly due to the paper's analysis and more ablation studies. In this rebuttal phase, we will try to address them carefully, and we would be grateful if you could update your score if our responses resolve your concerns.**

---

### Author Response · Authors · 2025-11-28

## **Paper Revision**
We thank all reviewers for the constructive feedback. Here we summarize the revision of the manuscript according to the comments and suggestions of reviewers:

- In sections 3 and 5, we revise the instance according to the reviewer's suggestion.
- In section 3.3, we include detailed explanations on dataset validation on caption diversity and alignment accuracy.
- In section 4.3, we provide the details on the random drop cfg training.
- In section 5.2, we include more training details as well as the synchronization feedback for a clearer presentation.
- In section 6.1, we carefully cite and provide additional evaluation with 1) Tell What You Hear From What You See, and 2) FoleyGen baselines and put ReasonAudio's video-to-audio ablations in Appendix C.
- In section 6.2, we provide reasoning baselines with ThinkSound trained on VGGSound-Think, ensuring that the text conditioning is consistent with the ReasonAudio training procedure.
- In section 6.3, we attach the results with the hybrid CLIP and LLM model, and the model with the Omni MLLM backbone.
- In section 6.3, we provide more explanation on audio quality and synchronization trade-off in DPO post-training.
- In Appendix D, we add a new subsection to include GPT-4o prompting, GPT-4o comparison with open-source LLMs, and discuss the differences in text format between VGGSound-Think and VGGSound.
- In Appendix F, we ablate the number of learnable queries for the MLLM understanding module.
- In Appendix G, we include more details on human evaluation.
- In Appendix H, we evaluate model generalization on the MovieGen Audio benchmark and report objective metrics.

To address and clarify these concerns, we carefully run new experimental results with different requirements, modify our manuscripts, and provide responses to your questions and comments. We are always happy to have a further discussion and answer more questions raised by you. Thanks in advance,

Paper 2216 authors

---

### Author Response · Authors · 2025-12-04

# **Summary of the response (1/2)**

To all reviewers, ACs, and PCs:
We thank all reviewers for the valuable suggestions with their effort and time, and for considering our work in positive contributions:

- **Reviewer 6u3X. "key contribution highly original", "VGGSound-Think is a high quality contribution", and "clearly addresses two major challenges".**
- **Reviewer 2MVJ. Be happy to discuss. "use of LLM brings insightful capabilities", "quantitative and qualitative improvements", and "clear and polished writing"**
- **Reviewer D3zi. "two real pinpoints, proposed method is clear", "$\beta$ ablations are helpful'', and "proposed datasets are valuable".**
- **Reviewer 3J1P. Be willing to raise scores. "paper is written well enough", "methods sound reasonable", and "experiments are comprehensive, and results are good".**

Your comments have improved our work. Here we summarize our efforts in addressing reviewers' concerns, and please refer to the responses to each reviewer for more details. To address and clarify concerns, we run **extensive experimental comparison to demonstrate the effectiveness of proposed modules, provide more detailed explanations on dataset and model design, and present precise definitions and presentation**.

**[Extensive experimental comparison to demonstrate the effectiveness of proposed modules.]**

- Following the reviewers’ suggestions, we carefully run **comparisons with additional baselines (VATT-Gemma-T and Foleygen)**, and provide analysis on synchronization feedback and technical comparison (ThinkSound and MMAudio). The baselines with MLLM (Thinksound and VATT-Gemma-T [1]) necessitate tuning LLMs on video understanding and subsequently training the audio generator, and ReasonAudio uses learnable queries to directly bridge multimodal understanding and audio generation within a unified framework, strengthening end-to-end multimodal reasoning.

- Per the reviewers' suggestions, we include the **ablation studies with the hybrid pipeline that uses CLIP for visual understanding and an LLM for reasoning text**. The ablation highlights the advantage of MLLMs as the multimodal understanding component, which better integrates visual cues with textual reasoning, strengthening audio–visual reasoning, and generates audio semantically aligned with the input video.

- In response to the reviewer's question, we present the results of **video-to-audio synthesis (video-only mode) or text-to-audio synthesis (text-only mode)** without additional fine-tuning. ReasonAudio under pure V2A conditioning achieves strong cross-modal coherence, with the SOTA ImageBind and DeSync scores, while maintaining comparable perceptual quality (FD) to competitive baselines.


- Per the reviewers' suggestions, we include video-to-audio studies and compare with ThinkSound/MMAudio **on the MovieGen Audio benchmark to assess generalization**. The results highlight the advantage of using MLLMs as the understanding module: Compared to the CLIP encoder, they generalize more effectively and provide stronger semantic reasoning.

---

### Author Response · Authors · 2025-12-04

## **Summary of the response (2/2)**

**[More detailed explanations on dataset and model design.]**

- In response to the reviewer's question, we explain the **number of learnable queries** intended to enable a fair comparison between LLM and CLIP-based conditioning by matching the representation shape, and ablate the understanding module on the VGGSound-Think test set by varying the number of learnable queries.

- As shown in our responses, we present the **latency comparison** and find that the MLLM with learnable prompts produces conditioning latents in a single forward pass, significantly improving generation efficiency and avoiding LLM overfitting or multi-stage training.


- In response to the reviewer's question, we evaluate and **compare GPT-4o with VideoLLaMA3 as an open-source MLLM**, to extract temporal and semantic information from videos. It shows that this typical open-source model is less efficient at following our instruction format compared to GPT.

- For **dataset validation**, we use metrics (mean pairwise cosine distance, VLM-as-Judge) to **evaluate caption diversity and alignment accuracy**. VGGSound-Think exhibits distinctly higher caption diversity and alignment accuracy than VGGSound. In VGGSound, the text annotations are often sparse and semantically shallow.

- In response to the reviewer's question on **DPO trade-off analysis**, we explain in detail that $\beta$ is leveraged as a hyperparameter to control the trade-off between the strength of the policy update and distance to the pretrained model, the alignment–fidelity trade-off we observe persists in previous reward-finetuning works (tango, musicdpo).


**[More precise definitions and presentation.]**


- In response to the reviewer's question, we attach the **details of human evaluation**, as well as the random conditional feature dropping to text or video conditions independently.

- Per the reviewers' suggestions, we refine the mentioned **Qwen-2.5 instance** in the revised version of the paper.

- For RoPE, we explain the temporally **aligned RoPE** to the queries and keys of both the visual and audio streams (similar to MMAudio and ThinkSound).

- In response to the reviewer's question, we fix the **sample display** issue.

**In the meanwhile, we revise the manuscript according to the comments and suggestions of reviewers. In this rebuttal phase, we have tried to address them carefully, and we sincerely hope you find our replies address your concerns.**

**Best regards, Authors**

---

### Note · Authors · 2026-01-06

I have read and agree with the venue's withdrawal policy on behalf of myself and my co-authors.